

**Bio-optical characterization of subsurface chlorophyll maxima in**
**the Mediterranean Sea from a Biogeochemical-Argo float**
**database**
Marie Barbieux[1], Julia Uitz[1,] Bernard Gentili[1], Orens Pasqueron de Fommervault[2], Alexandre Mignot[3], Antoine Poteau[1],
Catherine Schmechtig[4], Vincent Taillandier[1], Edouard Leymarie[1], Christophe Penkerc'h[1], Fabrizio D'Ortenzio[1], Hervé
Claustre[1] & Annick Bricaud[1]
[1]CNRS and Sorbonne Université, Laboratoire d'Océanographie de Villefanche, LOV, F-06230 Villefranche-sur-mer, France
[2]Alseamar-alcen company, 9 Europarc Sainte Victoire,13590 Meyreuil, France
[3]Mercator Océan, 31520 Ramoville Saint Agne
[4]OSU Ecce Terra, UMS 3455, CNRS and Sorbonne Université, Paris 6, 4 place Jussieu 75252 Paris cedex 05, France
*Correspondence to*: Marie Barbieux (marie.barbieux@obs-vlfr.fr)
**ABSTRACT**
As commonly observed in oligotrophic stratified waters, a Subsurface (or Deep)
Chlorophyll Maximum (SCM) frequently characterizes the vertical distribution of
phytoplankton chlorophyll in the Mediterranean Sea. Occurring far from the surface layer
"seen" by ocean color satellites, SCMs are difficult to observe with adequate spatio-temporal
resolution and their biogeochemical impact remains unknown. BioGeochemical-Argo (BGC-
Argo) profiling floats represent appropriate tools for studying the dynamics of SCMs. Based
on data collected from 36 BGC-Argo floats deployed in the Mediterranean Sea, our study
aims to address two main questions: (1) What are the different types of SCMs in
Mediterranean Sea? (2) Which environmental factors control their occurrence and dynamics?
First, we analyzed the seasonal and regional variations of the chlorophyll concentration
(Chl$a$), particulate backscattering coefficient ($b_{bp}$), a proxy of the Particulate Organic Carbon
(POC), and environmental parameters (PAR and nitrates) within the SCM layer over the
Mediterranean basin. The vertical profiles of Chl$a$ and $b_{bp}$ were then statistically classified,
and the seasonal occurrence of each of the different types of SCMs quantified. Finally, a case
study was performed on two contrasted regions and the environmental conditions at depth
were further investigated to understand which parameter controls the SCMs. In the Eastern
Basin, SCMs result, at a first order, from photoacclimation process. Conversely, SCMs in the
Western Basin reflect a biomass increase at depth benefiting from both light and nitrate
resources. Our results also suggest that a variety of intermediate types of SCMs are
encountered between these two end-member situations.





## 1   INTRODUCTION

The vertical distribution of phytoplankton in the open ocean is often characterized by

the occurrence of high chlorophyll *a* concentration (Chl*a*) beneath the mixed layer (Cullen
and Eppley, 1981; Fasham et al., 1985; Raimbault et al., 1993; Letelier et al., 2004; Tripathy
et al., 2015). This phenomenon is commonly referred to as Deep Chlorophyll Maximum
(DCM) or Subsurface Chlorophyll Maximum (SCM). Although it always happens below the
surface layer (approximately below the first 20 meters), it does not necessarily settle very
deep in the water column, thus making the notation DCM sometimes inappropriate. Hence, in
the following, we will use the notation SCM. Commonly observed at depth in oligotrophic
stratified regions (Anderson, 1969; Cullen, 1982; Furuya, 1990; Mignot et al., 2014), SCMs
are also known to occur below the mixed layer in temperate- and high-latitude environments
(Parslow et al., 2001; Uitz et al., 2009; Ardyna et al., 2013; Arrigo et al., 2011). The
formation of a subsurface maximum of Chl*a* in these different ecosystems results from
various underlying mechanisms leading to different types of SCMs. In stratified waters,
SCMs often result from photoacclimation of the phytoplankton organisms, which induces an
increase in the intracellular Chl*a* in response to low light conditions (Dubinsky and Stambler,
2009; Fennel and Boss, 2003; Kiefer et al., 1976; Winn et al., 1995). However SCMs
resulting from an actual increase in phytoplankton carbon biomass have also been reported in
such ecosystems (Beckmann and Hense, 2007; Crombet et al, 2011; Mignot et al., 2014). In
high-latitude regions with well-mixed surface waters, SCMs have been shown to result from
the accumulation of particles sinking from the mixed layer (Quéguiner et al., 1997; Parslow et
al, 2001), photophysiological acclimation of algal cells (Mikaelyan and Belyaeva, 1995) or
phytoplankton growth at the depth of the nutricline (Holm-Hansen and Hewes, 2004; Tripathy
et al 2015). Hence, regional or local studies have highlighted underlying processes indicating
that, under certain conditions, SCMs could contribute to carbon production and export, and



thus potentially have an important biogeochemical role. However, we have limited knowledge
of their biogeochemical significance at large spatial and temporal scales. Their contribution to
the depth-integrated primary production has been assessed for a limited number of regions
and remains largely unknown, although it has been reported to be underestimated from 40 to
75% in the Arctic Ocean (Ardyna et al, 2013; Hill et al, 2013). The biogeochemical
contribution of the SCMs to the global ocean is also particularly hard to assess at large spatio-
temporal scales, especially because SCMs settle at a depth usually far from the surface layer
"seen" by ocean color satellites.  Remotely sensed estimates are restricted to the upper layer
of the water column that represent only one fifth of the euphotic layer where phytoplankton
photosynthesis takes place (Gordon and McCluney, 1975). The exact biogeochemical role of
SCMs, thus, needs to be further explored.

The Mediterranean Sea is considered as an oligotrophic province where the vertical

distribution of phytoplankton is, seasonally or permanently, characterized by the occurrence
of a SCM (Christaki, 2001; Estrada et al., 1993; Kimor et al., 1987; Lavigne et al., 2015;
Siokou-Frangou et al., 2010; Videau et al., 1994). It is also a low-nutrient concentration basin,
one of the largest nutrient-depleted areas of the global ocean and it is characterized by a west-
to-east gradient in both nutrients and chlorophyll $a$ concentration (Dugdale and Wilkerson,
1988; Bethoux et al., 1992; Antoine et al., 1995; Bosc et al., 2004; D'Ortenzio and Ribera
d'Alcalà, 2009). While the Eastern Basin is defined as oligotrophic (Krom et al., 1991;
Ignatiades et al., 2002; Lavigne et al., 2015), the Western Basin is more productive and
behaves as a temperate system (Morel and André, 1991; Marty et al., 2002; Mayot et al.,
2017b). Hence, this "miniature ocean" presents SCMs that may be encountered in both
temperate environments and stratified waters of the global ocean. This, coupled with an
intensive effort of biogeochemical observations in this region, makes the Mediterranean Sea
an ideal region for studying SCMs.





The biogeochemical and bio-optical community recently developed autonomous
profiling floats that collect *in situ* vertical profiles of biogeochemical properties such as the
chlorophyll *a* fluorescence (*i.e.* a proxy of the chlorophyll *a* concentration (Chl*a*)) and the
particulate backscattering coefficient ($b_{bp}$) (*i.e.* a proxy of the Particulate Organic Carbon
(POC)). Physical-chemical properties such as nitrate concentration ($[NO_3^-]$) or the
Photosynthetically Available Radiation (PAR), essential to understand the functioning of
SCMs, are also measured simultaneously (Johnson et al., 2009; Claustre et al., 2010; Johnson
and Claustre, 2016). Thirty-six BioGeochemical-Argo (BGC-Argo) have been deployed in the
Mediterranean Sea from 2012 to 2017, providing a database of 4050 *in situ* multi-variable
profiles. This extensive database gives us the unique opportunity to enhance our
comprehension of the vertical distribution and seasonal variability of the phytoplankton
biomass in the subsurface layer of the Mediterranean Sea and expand our understanding of the
mechanisms involved in the occurrence of SCMs. Our study seeks to address two main
questions: (1) What are the different types of SCMs in Mediterranean Sea?; and (2) Which
environmental factors control the occurrence and dynamics of the different types of SCMs in
this region? To address these questions, three complementary approaches were used. First,
based on a climatological approach, we analyzed the spatial and seasonal variability of
biogeochemical properties (*i.e.* Chl*a* and $b_{bp}$) and environmental conditions at the SCM level.
This should lead to the identification of the main mechanisms controlling SCMs in different
regions of the Med Sea. Second, using a statistical method, we classified the vertical profiles
of Chl*a* and $b_{bp}$ seasonally encountered in the various regions of the Med. This approach
allowed us to quantify the frequency of occurrence of distinct types of SCMs in these
different regions. Finally, using two specific BGC-Argo floats deployed in the Gulf of Lions
and the Levantine Sea, we conducted a case study of two contrasted regimes and investigate
the environmental conditions that control the occurrence of SCMs in each regime.



## 2 DATA AND METHODS

### 2.1 The BGC-Argo profiling float database

Thirty-six BGC-Argo profiling floats were deployed in the Mediterranean Sea in 5 geographic areas, *i.e.* the Northwestern (NW) and Southwestern (SW) regions, the Tyrrhenian (TYR), Ionian (ION) and Levantine (LEV) Seas. Our study was based on the analysis of a database comprising 4050 multivariable vertical profiles, corresponding to upward casts collected between November 26, 2012 and September 27, 2017 (Table 1 and Figure 1). The ''PROVOR CTS-4'' (NKE Marine Electronics, Inc.) is a profiling autonomous platform that has been specifically designed in the frame of the remOcean and NAOS projects. The physical variables (depth, temperature and salinity) were acquired by a SBE 41 CTD (Sea-Bird Scientific Inc.). Two optical packages, the so-called remA and remB, were developed to be specifically implemented on profiling floats. The remA is composed of an OCR-504 (SAtlantic, Inc.), a multispectral radiometer that measures the Photosynthetically Available Radiation (PAR) and the downwelling irradiance at 380, 410 and 490 nm. The remA also includes an ECO3 sensor (Combined Three Channel Sensors; WET Labs, Inc.) measuring the fluorescence of the chlorophyll *a* and the Colored Dissolved Organic Matter (CDOM) at excitation/emission wavelengths of 470/695 nm and 370/460 nm, respectively, and the angular scattering coefficient of particles ($\beta(\theta, \lambda)$) at 700 nm and at an angle of 124°. Finally, 15 floats were also equipped with a nitrate ($NO_3^-$) (Deep SUNA, Sea-Bird Scientific, Inc.) or/and an oxygen ($O_2$) sensor (optode 4330, Aanderaa, Inc.). Depending on the scientific objectives of the different projects, the measurements were collected during upward casts programmed every 1, 2, 3, 5, or 10 days. All casts started from a parking depth at 1000 m at a time that was sufficient for surfacing around local noon. The vertical resolution of data acquisition was 10 m between 1000 m and 250 m, 1 m between 250 m and 10 m, and 0.2 m



between 10 m and the surface. Each time the floats surfaced, the raw data were transmitted to
land through Iridium two-way communication.

## 2.2   Retrieval of key biogeochemical variables from optical measurements

For each bio-optical parameter, raw counts were converted into the desired quantities

according to technical specifications and calibration coefficients provided by the
manufacturer. These quantities were transformed into chlorophyll *a* concentration (Chl*a*) and
particulate backscattering coefficient ($b_{bp}$) following the BGC-Argo procedure (Schmechtig et
al., 2015, 2016b). In addition, we applied a global factor of 2 to correct chlorophyll a
fluorescence measurements from WET Labs ECO fluorometers, following the
recommendation of Roesler et al. (2017). This correction factor applied to BGC-Argo data
was found to have little impact on the interpretation of the results on a global scale (Barbieux
et al., 2018; Organelli et al., 2017) and did not modify the interpretation of the present results,
especially because the regional correction factors proposed by Roesler et al. (2017) for the
Mediterranean Sea are very close to the global factor of 2 (1.62 and 1.72 for the Western and
Eastern Basin, respectively). Finally a quality-controlled procedure was performed following
the BGC-Argo recommendations (Schmechtig et al., 2016a). All data were also visually
checked in order to detect any drift over time or sensor deficiency. These data were made
freely available by the International Argo Program (http:// www.argo.ucsd.edu,
http://argo.jcommops.org) and the CORIOLIS project (http://www.coriolis.eu.org).

After binning the data at a 1-m resolution, the mixed layer depth (MLD) was derived

from the CTD data using a 0.03 kg m$^{-3}$ density criterion (de Boyer Montégut, 2004). The
depth of the SCM and of the Subsurface $b_{bp}$ Maximum (S$b_{bp}$M) was identified as the depth
where the absolute value of Chl*a* or $b_{bp}$ reaches a maximum below the MLD. Large spikes
associated with particle aggregates or zooplankton (Gardner et al., 2000; Briggs et al., 2011)





were observed in the $b_{bp}$ profiles and made it sometimes difficult to identify the depth of the
$Sb_{bp}M$. Hence, for the purpose of the $Sb_{bp}M$ retrieval exclusively, the $b_{bp}$ values were
smoothed with a mean filter (5-point window). To study the SCM dynamics and obtain the
width of the SCM that may fluctuate in space and time, a Gaussian profile was adjusted to
each Chl*a* vertical profile of the database that presented a SCM. This approach first proposed
by Lewis et al. (1983) has been widely used in oceanographic studies (*e.g.* Morel and
Berthon, 1989; Uitz et al., 2006; Barbieux et al., 2017). The width of the gaussian adjusted to
the vertical profile of Chl*a* represented the width of the SCM. The SCM layer was defined as
the layer extending across the entire width of the SCM. The upper (or lower) limit was
retrieved by removing (or adding) half of the width of the SCM to the absolute depth of the
SCM.

### 2.3    Estimation of nitrate concentration

The SUNA sensor measures the light absorption in the wavelength range from 217 to
240 nm. In this spectral band, the light absorption is dominated by nitrates and bromides, and,
to a much lesser extent, by organic matter (Johnson and Coletti, 2002). Various algorithms
were developed to obtain the nitrate concentration ($[NO_3^-]$) from the measured light
absorption spectrum (*e.g.* Arai et al., 2008; Zielinski et al., 2011). The TCSS algorithm was
specifically developed to take into account the temperature dependency of the bromide
spectrum, which significantly improved the accuracy of the retrieved $[NO_3^-]$ (Sakamoto et al.,
2009). This algorithm was recently modified to also take into account a pressure dependency
(Pasqueron de Fommervault et al., 2015a; Sakamoto et al., 2017). Previous studies also
evidenced the inaccuracy of standard calibration procedures (D'Ortenzio et al., 2014;
Pasqueron de Fommervault et al., 2015a) and showed that SUNA sensors often undergo offset
issue and drift over time (Johnson and Coletti, 2002). Johnson et al. (2017) proposed a
method to correct these issues for the Southern Ocean. Using the GLODAP-V2 database





(http://cdiac.ornl.gov/oceans/GLODAPv2) of *in situ* measurements, the authors determined an
empirical relationship allowing the estimation of the [$NO_3^-$] at depth ([$NO_3^-$]$_{deep\_pred}$  for
nitrate concentration deep reference value) using a multiple linear regression (MLR) with
physical and geolocation parameters as predictors (salinity, temperature, oxygen, latitude and
longitude). BGC-Argo profiles of nitrate concentration were then corrected by adjusting the
SUNA measurements to the retrieved deep reference value. Following a similar approach, we
established a regional empirical relationship for the Mediterranean Sea (Eq.1) allowing to
retrieve the [$NO_3^-$]$_{deep\_pred}$  values  using parameters that were systematically measured by
the BGC-Argo floats (*i.e.* latitude, longitude, temperature and salinity). For the Mediterranean
Sea, oxygen was not used as an input parameter of the MLR as this parameter was not
systematically available for the BGC-Argo floats of our database. Moreover, its absence in
the MLR as an input parameter did not affect the retrieval of the nitrate concentrations.
Comparing the nitrate concentrations predicted by the MLR to the nitrate concentrations from
GLODAP-V2 data, the determination coefficients of the relationship presented very similar
values for the model with and without oxygen (see Figure S1 in Supplement 1).
Hence, the following equation was finally used:
[$NO_3^-$]$_{deep\_pred}$ = 454.28 – 0.002 x Latitude – 0.0473 x Longitude + 1.7262 x Temperature –
12.165 x Salinity                                                              (1)
A strong correlation was noticed between the nitrate concentrations predicted from the MLR
model and the measurements provided in the GLODAP-V2 database. This correlation was
associated with a strong determination coefficient ($R^2$ = 0.89) and a small root mean square
error (RMSE = 0.52 µmol L$^{-1}$). Then, comparing the predicted climatology-based with the



observed BGC-Argo nitrate concentrations at depth and computing the adjusted nitrate
concentration for each depth, we obtained the following equation:
$[NO_3^-]_{adjusted} (t,z) = [NO_3^-]_{raw} (t,z) - ([NO_3^-]_{deep\_obs} (t) - [NO_3^-]_{deep\_pred} (t))$       (2)
with $[NO_3^-]_{raw} (t,z)$ corresponding to the raw nitrate value from the SUNA sensor.

The BGC-Argo $[NO_3^-]$ profiles of the Mediterranean database were compared with *in*

*situ* measurements collected simultaneously to float deployment (see Taillandier et al., 2017
for more details), using the classic colorimetric method (Morris and Riley, 1963). We
demonstrated that the retrieval of the BGC-Argo $[NO_3^-]$ with the proposed calibration
procedure was satisfying. The comparison between the nitrate concentrations retrieved from
the BGC-Argo floats to the reference *in situ* measurements (Figure 2) showed a robust
relationship ($R^2 = 0.86$ and slope = 0.97, N = 162).

The nitracline that separates upper nitrate-depleted waters from lower repleted waters

corresponds, in this paper, to the depth where $[NO_3^-]$ is 1 µM smaller than the median $[NO_3^-]$
value in the first 10 meters of the water column (Lavigne et al., 2013). The diffusive vertical
supply of nitrates to the euphotic zone is not only influenced by the depth of the nitracline
from the sunlit surface layer but also by the slope of the nitracline. The slope of the nitracline
was calculated as the vertical $[NO_3^-]$ gradient between the isocline 1 µM and the isocline 3
µM as already done for the Mediterranean Sea by Pasqueron de Fommervault et al. (2015a).
**2.4   Estimation of daily PAR**
The BGC-Argo vertical profiles of PAR were quality-checked following Organelli et al.
(2016). Only solar noon profiles were considered for our analysis because zenith
measurements ensure the best retrieval (Organelli et al., 2017) of the isolume, *i.e.* depth



corresponding to a chosen value of light. BGC-Argo floats provide instantaneous PAR
(iPAR) measurements just beneath the sea surface at local noon (iPAR($0^-$, noon)).

From iPAR measurement, a vertical profile of daily-averaged PAR was estimated

following the method of Mignot et al. (2018). This method relies on a theoretical clear-sky
estimate of iPAR just beneath the sea surface using the solar irradiance model SOLPOS
developed by the National Renewable Energy Laboratory (NREL, 2000). Hence, we followed
three main steps:
(1) The instantaneous photosynthetically available radiation just beneath the sea surface at
time t, iPAR($0^-$, t) in µmol photons m$^{-2}$ s$^{-1}$, was determined from Eq. (3):
$$iPAR\ (0^-, t) = \ iPAR_{clear}\ (0^-, t) \frac{iPAR\ (0^-, noon)}{iPAR_{clear}\ (0^-, noon)}$$    (3)
with iPAR$_{clear}$ $(0^-, t)$ the theoretical estimate of iPAR just beneath the sea surface at time t,
iPAR($0^-$, noon) the float measurement of iPAR just beneath the sea surface at local noon,
and iPAR $_{clear}(0^-, noon)$ the theoretical estimate of iPAR just beneath the sea surface at local
noon for the same time and location as the float measurement. The ratio of iPAR($0^-$, noon) to
iPAR$_{clear}(0^-, noon)$ represented an index of the cloud coverage at noon, which was applied
to the clear-sky iPAR estimates at any time t. This approach thus assumes that the cloud
coverage at noon is representative of the daily cloud coverage. Although the cloud coverage is
unlikely to be constant throughout the entire day, this approach enabled to account for the
daily course of light through modeled estimates, rather than considering only the noon-time
instantaneous float measurements.
(2) The daily-averaged PAR just beneath sea surface, PAR(0−) in mol photons m$^{-2}$ d$^{-1}$, was
obtained by averaging Eq. (3) over daylength. In parallel, the diffuse attenuation coefficient
for PAR, K$_d$(PAR) in unit of m$^{-1}$, was derived from the float iPAR measurements by fitting a
linear    least    square    regression    forced    through    the    origin    between    the    data    of



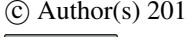

$\ln \left( \frac{\text{iPAR}_{\text{float}}(z, \ \text{noon})}{\text{iPAR}_{\text{float}}(0^{-},\text{noon})} \right)$ and z taken in the upper 40 m of the water column (Mignot et al.,

2018).

(3) Finally, the daily-averaged PAR for each depth z of the water column, PAR(z) in units of
mol photons $m^{-2}$ $d^{-1}$, was calculated from $K_d$(PAR) and PAR(0−) as follows:
$\text{PAR}\,(z) = \text{PAR}(0^{-})\,{}_{\exp\,(\,K_d(\text{PAR})z\,)}$                    (4)

Additionally, the isolume 0.3 mol quanta.$m^{-2}$ $d^{-1}$, which corresponds to the dataset

median daily PAR value at the SCM depth, was used as an indicator of the light available for
photosynthesis at the SCM level. We also computed the euphotic layer depth ($Z_{eu}$) as the
depth where the PAR is reduced to 1% of its surface value (Gordon and McCluney, 1975) and
the penetration depth ($Z_{pd}$) calculated as $Z_{eu}$ / 4.6. The surface layer corresponds to the layer
extending from 0 to $Z_{pd}$.
**2.5   Statistical method of classification of the vertical profiles providing the**

**identification of the SCM**

A statistical method based on the Singular Value Decomposition (SVD) algorithm

(Golub and Van Loan, 1996) was used to identify the different types of SCMs in the
Mediterranean Sea. The approach allowed to tackle the large amount of data provided by the
BGC-Argo floats and to simultaneously classify the Chl*a* and $b_{bp}$ vertical profiles of the
database. Based on the shape of the Chl*a* or $b_{bp}$ vertical profile, the method did not require an
*a priori* knowledge on the considered profile such as in previous studies (*e.g.* Uitz et al. 2006;
Mignot et al., 2011; Lavigne et al., 2015). The present method involved three major steps
summarized as follows (see Supplementary Material 2 for more details):
(1) Each vertical profile of Chl*a* and $b_{bp}$ were normalized in depth and magnitude. The depths
were normalized by the euphotic depth ($Z_{eu}$) and the Chl*a* and $b_{bp}$ values were normalized to



the maximum value of each profile (i.e. Chl$a_{max}$ and $b_{bpmax}$, respectively). Ultimately, the
Chl$a$ and $b_{bp}$ values of a profile were joined by one end, to obtain a dimensionless, twice as
long, "metaprofile" that was subsequently classified on the basis of its shape.
(2) A Principal Component Analysis (PCA) was performed using the Singular Value
Decomposition algorithm (Pearson, 1901). The singular values were ordered in decreasing
order and only the first N values were kept. N was chosen so that the corresponding singular
vectors capture 95% of variance of the dataset and the resulting vertical profiles of Chl$a$ and
$b_{bp}$ were ecologically meaningful (see Supplement 2 provided as electronic supplementary
material).
(3) Each singular vector defined a profile shape. A dimensionless metaprofile can be
represented as a linear combination of those shapes, each multiplied by a coefficient. To
classify each metaprofile in a category of shape, we used a numerical optimization algorithm
on the whole set of coefficients to maximize the value of one coefficient while minimizing the
N-1 others, for each metaprofile. The coefficient that was maximal for each metaprofile
defined its class of shape. More details on the method are provided as electronic
supplementary material.

For each of the five regions of the Mediterranean considered, we finally obtained the

dominant shapes of vertical Chl$a$ and $b_{bp}$ profiles, which are representative of the different
situations encountered along an annual cycle. This approach allowed to establish a typology
of SCMs in the BGC-Argo database and to report their frequency of occurrence in each
region.



## 3    RESULTS & DISCUSSION

### 3.1    Regional and seasonal variability of the SCM

Using a climatological approach, we first examined the characteristics of the SCMs such as their depth, thickness and amplitude in order to better apprehend their vertical dynamics in the water column along the Mediterranean west-to-east gradient. Then, the seasonal variations of the biogeochemical properties (Chl$a$ and $b_{bp}$) at the SCM level were studied in relation to environmental conditions. This ultimately leaded us to identify and describe the main types of SCMs in the five considered regions of the Med Sea.

### 3.1.1    Variability of the SCM along the west-to-east gradient

The well-known west-to-east trophic gradient of the Mediterranean was observed in the present dataset, with a decrease in the surface Chl$a$ from the NW region (median value of 0.15 mg m$^{-3}$) to the LEV region (median value of 0.04 mg m$^{-3}$; Figure 3a). A decrease in the amplitude of the SCM paralleled the surface gradient, with decreasing mean Chl$a$ and $b_{bp}$ values in the SCM from the NW to the LEV (0.45 to 0.24 mg m$^{-3}$ and 0.00088 to 0.00050 m$^{-1}$ for Chl$a$ and $b_{bp}$, respectively) (Figures 3b-c). In the Eastern Basin (*i.e.* ION and LEV), only 27% of the Chl$a$ values were distributed above the median value calculated for the entire Mediterranean Basin (0.28 mg m$^{-3}$) whereas 66% of the Chl$a$ values exceeded it in the Western Basin (*i.e.* NW, SW, and TYR; Figure 4). Similarly, in the Eastern Basin, only ~30% of the $b_{bp}$ values exceeded the median value calculated for the entire Mediterranean Sea in the SCM (0.00058 m$^{-1}$) (*i.e.* 32% and 29% for the ION and LEV, respectively; Figures 4d-e) whereas in the Western Basin, ~75% of the $b_{bp}$ values were distributed above the global median value (*i.e.* 81%, 80% and 71% for the NW, SW and TYR, respectively, Figures 4a-c).



In parallel, from the NW to the LEV regions, a deepening of the SCM (median values
of 58 and 95 m, respectively; Figure 3d) and an increase in its thickness (median values of 43
and 72 m, respectively; Figure 3e) was observed. A statistical Wilcoxon test revealed non-
identical distributions of the considered variables (SCM amplitude, depth and thickness)
among the different Mediterranean regions (significance level p < 0.001). Our results suggest
that the well-known west-to-east trophic gradient of the Mediterranean occurs not only at the
surface but also at depth. As suggested by previous studies (Mignot et al., 2014; Lavigne et
al., 2015), we confirm that the thickness and depth of the SCM are inversely related to its
amplitude. The eastward weakening, deepening and increase in the thickness of the SCM is
gradual across the Mediterranean Sea.
**3.1.2   Seasonal variations of Chl$a$ and $b_{bp}$**
The seasonal cycle of the Chl$a$ in the SCM was more pronounced in the Western
Basin than in the Eastern Basin. This was especially true for the NW (Figure 4a) with median
values of Chl$a$ reaching ~0.8 mg m$^{-3}$ in June-July and ~0.3 mg m$^{-3}$ in January-February.
Similarly, the seasonal cycle of $b_{bp}$ in the SCM was more pronounced in the occidental part of
the Med Sea than in the oriental part. Depending on the region and period of the year, the
Chl$a$ and $b_{bp}$ values showed synchronous or decoupled seasonal cycles. In the Western Basin,
the $b_{bp}$ and Chl$a$ seasonal cycles were coupled. The NW and TYR regions of the Western
Basin showed a seasonal cycle characterized by two Chl$a$ peaks at the SCM in March-April
and June-July (the SW region presents a single maximum from April to July) and a
simultaneous increase in $b_{bp}$ recorded in April-June (Figures 4a-c). On the opposite, the ION
and the LEV presented a unique maximum of Chl$a$ in June that is delayed compared to the $b_{bp}$
seasonal maximum occurring in February-April (Figures 4d-e).





The chlorophyll *a* concentration is the most commonly used, yet imperfect, indicator of

the phytoplankton biomass (Cleveland et al., 1989; Geider, 1993). Variations in Chl*a* may
reflect changes in either phytoplankton carbon (Furuya, 1990; Hodges and Rudnick, 2004;
Beckmann and Hense, 2007) or in intracellular content as a result of physiological processes
occurring in phytoplankton cells, photoacclimation in particular (Geider et al., 1997; Fennel
and Boss, 2003). The particulate backscattering coefficient is considered as a proxy of the
abundance of particles (Morel and Ahn, 1991; Stramski and Kiefer, 1991; Loisel and Morel,
1998; Stramski et al., 2004) and of the stock of Particulate Organic Carbon (POC) in the open
ocean waters (Stramski et al., 1999; Balch et al., 2001; Cetinić et al., 2012; Dall'Olmo and
Mork, 2014). In contrast with Chl*a*, it provides information on the whole pool of particles, not
specifically on phototrophic organisms. The backscattering coefficient also depends on
several parameters such as the size distribution, nature, shape, structure and refractive index
of the particles (Morel and Bricaud, 1986; Babin and Morel, 2003; Huot et al., 2007b;
Whitmire et al., 2010).

The vertical and seasonal coupling of Chl*a* and $b_{bp}$ has been shown to reflect an actual

increase in carbon biomass whereas a decoupling could result from photoacclimation or from
a change in the nature or size distribution of the particle assemblage (Behrenfeld et al., 2005;
Flory et al., 2004; Siegel et al., 2005). The results presented above indicate that the Western
Basin presents higher values of Chl*a* and $b_{bp}$ in the SCM compared to the Eastern Basin and
displays a coupling of the properties all year long (Figure 4). Hence, we suggest that in the
NW, SW and TYR regions, the SCM sustains larger phytoplankton carbon biomass than in
the ION and LEV regions. Furthermore, in this Eastern part of the Med Sea, the SCM results,
at first order, from physiological acclimation to low light and/or from a modification of the
nature of the particle assemblage. In the next section, we will analyse the environmental



conditions occurring at the SCM level and attempt to determine the factors underpinning the
seasonal occurrence of SCMs in the different regions.

### 3.1.3 Environmental factors controlling the SCM

From a bottom-up perspective, it is the balance between light and nutrient limitations
that influences the establishment of phytoplankton communities at depth (Kiefer et al., 1976;
Cullen, 1982; Klausmeier and Litchman, 2001; Ryabov, 2012; Latasa et al., 2016). To explore
the light-nutrient regime in the SCM, a monthly climatology of the isolume and nitracline in
the different considered regions was represented with the depth of the Subsurface Chl$a$ and
$b_{bp}$ Maxima (*i.e.* SCM and S$b_{bp}$M, respectively; Figure 5).
In the Western Basin, the isolume 0.3 mol quanta m$^{-2}$ d$^{-1}$, the nitracline 1 µmol, the
S$b_{bp}$M and the SCM were all located at a similar depth during the oligotrophic period
(maximum depth difference < 20 m; Figures 5a-c). Hence, this part of the Med Sea benefits
from both light and nutrient resources and presents an actual increase in phytoplankton
biomass (Figures 5 and 6a-b). The shallowest nitracline (median of 61 m; Figure 6c) and the
steepest nitracline (slope of 90 µmol m$^{-4}$; Figure 6d) were here recorded for the NW, which
may indicate higher upward diffusive flux of nitrates available to sustain phytoplankton
biomass.
In contrast, in the ION and LEV regions, the isolume 0.3 mol quanta m$^{-2}$ d$^{-1}$, nitracline 1
µmol, SCM and S$b_{bp}$M were not collocated in the water column (Figures 5d-e). The
difference between the depths of the SCM and nitracline was ~50 m during the stratified
period (Figures 5d-e and 6a) and the S$b_{bp}$M was shallower than the SCM (by ~40 m),
suggesting no accumulation of carbon at the SCM. In the Eastern Basin, the nitracline was
deeper (~120 m in Eastern Basin versus ~70 m in Western Basin; Figure 6c) and the nutrient
gradient was less sharp (nitracline slope of ~40 µmol m$^{-4}$ in Eastern Basin versus ~90 µmol m$^{-}$



[4] in Western Basin; Figure 6d) than in the Western Basin, suggesting a weak upward diffusive
flux of nitrates that corroborates previous results (Tanhua et al., 2013; Pasqueron de
Fommervault et al., 2015b). The inverse relationship between the nitracline steepness and the
thickness of the SCM is also confirmed (Gong et al., 2017). The PAR at the SCM level was
significantly lower in this Eastern part than in the Western part of the Med Sea (Wilcoxon test
at a significance level of $p < 0.001$; Figure 6b). The development of the SCM in this system
is, thus, limited by both the availability of light and nutrients. The SCM still settles at a depth
where light is available at a sufficient level to sustain photosynthesis, but never reaches the
nitracline.
**3.1.4   Coupling and decoupling of $b_{bp}$ and Chl$a$ in the SCM**
From the previous section, we have seen that the SCM of the Western Basin benefits
from both light and nutrient resources. In these conditions, the observed simultaneous
increase in Chl$a$ and $b_{bp}$ at the SCM most likely represents an actual development of
phytoplankton biomass, as indicated by the concordance between the depths of the SCM and
the S$b_{bp}$M (Figure 5). On the opposite, in the Eastern part of the Mediterranean, there is a
decoupling of Chl$a$ and $b_{bp}$ in the SCM, the maxima of Chl$a$ and $b_{bp}$ are not co-located. This
result suggests that environmental conditions, typically the light conditions, might inhibit the
increase in phytoplankton biomass. The microorganisms are, most probably, acclimated or
even adapted to these conditions. SCM species are indeed known to use different strategies
such as photoacclimation to low light (*i.e.* increase in the intracellular pigment content),
mixotrophy or small-scale directed movements towards light (Falkowski and Laroche, 1991;
Geider et al., 1997; Clegg et al., 2012). A vertical shift toward species more adapted to the
particular environmental conditions prevailing in the SCM layer is a well-known phenomenon
(*e.g.* Pollehne et al., 1993; Latasa et al., 2016). For example, two ecotypes of
*Prochlorococcus*, characterized by different accessory pigment contents, are known to be



adapted to either low-light or high-light conditions and to occupy different niches in the water
column (Bouman et al., 2006; Garczarek et al., 2007; Moore and Chisholm, 1999). In
particular, the low-light ecotype, characterized by increased intracellular pigmentation, has
been frequently observed at the SCM level in the Mediterranean, especially in the oriental
part (Brunet et al., 2006; Siokou-Frangou et al., 2010). A west-to-east modification in the
composition of phytoplankton communities in the SCM toward a dominance of
picophytoplankton species adapted to recurring light limitation, has been observed (Christaki,
2001; Crombet et al., 2011; Siokou-Frangou et al., 2010).
Whereas photoacclimation is defined as a short-term acclimation of a photosynthetic
organism to changing irradiance, photoadaptation refers to the long-term evolutionary
adaptation of photosynthetic organisms to ambient light conditions, through genetic selection.
Both phenomena could explain the vertical decoupling of $b_{bp}$ and Chl$a$ we observed in the
Eastern Basin. Yet our dataset did not allow us to conclude on the dominance of one process
compared to the other.
Although photoacclimation seems to be a prevalent hypothesis in numerous studies to
explain the vertical decoupling of Chl$a$ and $b_{bp}$ (*e.g.* Brunet et al., 2006; Cullen, 1982; Mignot
et al., 2014), it should yet be reminded that this decoupling could also result from a change in
the nature or size distribution of the particle assemblage. Small particles are, for example,
known to backscatter light more efficiently than large particles (Morel and Bricaud, 1986;
Stramski et al., 2004). A higher proportion of nonalgal particles in the Eastern compared to
the Western Basin could thus explain the decoupling between $b_{bp}$ and Chl$a$. The nonalgal
particles compartment is defined as the background of submicronic living biological cells (*i.e.*
viruses or bacteria) and non-living particles (*i.e.* detritus or inorganic particles) and is
typically known to represent a significant part of the particulate assemblage in oligotrophic
ecosystems (Claustre et al., 1999; Morel and Ahn, 1991; Stramski et al., 2001).





### 3.2 Classification of the Chl$a$ and b$_{bp}$ vertical profiles


In the previous section, we identified the major environmental factors leading to the
occurrence of two main types of SCMs in the five considered regions of the Med Sea. While a
concomitant maximum of Chl$a$ and b$_{bp}$ suggested a carbon biomass maximum, a decoupling
between the vertical distributions of these two properties may reflect photoacclimation, a
modification of the algal community composition, or a change in the nature and/or size of the
particle assemblage. The seasonal and regional variability in this global picture of the SCM
was explored using a statistical approach applied to the BGC-Argo dataset. Our aim was here
to classify the Chl$a$ and b$_{bp}$ profiles based on their shape. This leaded us to propose a typology
of the different types of SCMs seasonally encountered in the five regions of the
Mediterranean Sea. It also permitted to assess the frequency of these different types of SCMs
over the seasonal cycle and compare their characteristics among the various regions of the
Mediterranean Sea.

### 3.2.1 The NW: a region with a specific trophic regime


In the NW, the vertical distributions of Chl$a$ and b$_{bp}$ presented four different shapes
over the annual cycle (Figures 7a-b). The *mixed* shape was characterized by an homogeneous
distribution of Chl$a$ and b$_{bp}$ and showed occurrence exceeding 60% from December to March
(Figure 8a). The *bloom* shape exhibited high Chl$a$ and b$_{bp}$ values at surface with maximum
occurrence > 55% in April. The coexistence of the *mixed* and the *bloom* shapes during winter
and spring could result from intermittent mixing that alters the vertical distribution of Chl$a$
and b$_{bp}$ (*e.g.* Chiswell, 2011; Lacour et al., 2017). The *SBM$_{aZeu}$* and the *SBM$_{bZeu}$* (SBM
occurring above and below the euphotic depth, respectively) constituted two different cases of
subsurface maximum. In both cases, Chl$a$ and b$_{bp}$ covaried (Figures 7a-b), the maxima of



Chl$a$ and $b_{bp}$ were observed at nearly the same depth suggesting an increase in carbon
biomass in subsurface.

The $SBM_{aZeu}$ was often observed in late spring and late summer whereas the $SBM_{bZeu}$

occured more frequently (> 50%) in the middle of the oligotrophic period. This results
suggests a deepening of the SCM along the oligotrophic season and corroborates the "light-
driven hypothesis" previously formulated by Letelier et al. (2004) and Mignot et al. (2014). In
the NW region, the high surface Chl$a$ of the *bloom* shape (Figure 7a) probably results in
increased light attenuation in the water column from fall to spring. Consequently, the SCM
established shallower in spring than in summer (Figure 5a) and the $SBM_{aZeu}$ shape occurred
relatively frequently in spring (Figure 8a). Then, from spring to summer, the Chl$a$ decrease in
the surface layer of the water column resulted in decreased light attenuation and subsequent
deepening of the SCM (Figure 5a), which thus formed a subsurface maximum of Chl$a$ and $b_{bp}$
below the euphotic layer *($SBM_{bZeu}$*, Figure 8a). Therefore, our results are consistent with
previous studies (*e.g.* Gutiérrez-Rodríguez et al., 2010; Mayot et al., 2017b) that highlighted
the special status of the Northwestern region, the only region to exhibit the *bloom* shape and
predominantly SBMs during the oligotrophic season (Figures 9a-b).
**3.2.2    The SW and the TYR: regions of transition**

In the Southwestern region as well as in the Tyrrhenian Sea, three shapes characterized

the seasonal variability of the vertical distribution of Chl$a$ and $b_{bp}$ (Figures 7c-d and e-f). A
*mixed* shape, similar to that observed in the NW (Figures 9c-d), a *SBM* shape (Figures 9e-f),
and a *SCM* shape (decoupling between the maximum of Chl$a$ and $b_{bp}$ at depth) were
successively encountered over the seasonal cycle, with weak differences in their frequency of
occurrence among the two regions. The *SCM* shape established shallower in the water column
than the *SBM* shape (Figures 7c-f). It was encountered mainly in winter and fall (~50% of



occurrence), alternating with the *mixed* shape (Figures 8b-c). Thus, this shape probably
illustrates the erosion of the SCM by the winter mixing as previously suggested, for example,
in Lavigne et al. (2015). The *SBM* shape occurred mainly during spring and summer (>75%)
when both light and nutrients were available for phytoplankton growth (Figures 5b-c). The
*SBM* shapes of the SW and the TYR were comparable to the $SBM_{bZeu}$ shape of the NW
occurring at almost the same depth (~$Z_{eu}$). The *SCM* shapes of the SW and TYR were
analogous to the $SCM_{aZeu}$ shape of the ION and LEV (Figures 9e-h). Hence, our results
suggest that the SW and TYR regions are transition regimes that present types of SCMs that
can be found in both the Western and Eastern Basins.
**3.2.3    The ION and the LEV: oligotrophic end-members**
In the Ionian Sea, three different shapes were retrieved along the seasonal cycle, *i.e.* the
*mixed*, the $SCM_{aZeu}$ and the $SCM_{bZeu}$ shapes (Figures 7g-h). In this region, the Chl*a* maximum
was always decorrelated from the $b_{bp}$ maximum that revealed higher values at surface than at
depth. In the Levantine Sea, only two distinct shapes were encountered, *i.e.* the $SCM_{aZeu}$ and
the $SCM_{bZeu}$ shapes (Figures 7i-j). The subsurface maximum of Chl*a* was never associated
with a subsurface maximum of $b_{bp}$. Such SCMs constituted a permanent pattern with $SCM_{bZeu}$
and $SCM_{aZeu}$ reaching occurrences of 100% in June-July and > 75% in December-March,
respectively (Figures 8d-e). The $SCM_{bZeu}$ shape was a particularity of the Eastern Basin. This
shape was very similar in the ION and LEV, but very different from the shapes observed in
the other regions (Figures 9g-h). This $SCM_{bZeu}$ settled below the $Z_{eu}$ that, in such oligotrophic
systems, occurs relatively deep in the water column (~95 m; Figure 3d). This type of SCM
was also very thick (~70 m) (Figure 3e) and associated with low values of the nitracline slope
(Figure 6d).



### 3.3 A case study of the Gulf of Lions and Levantine Sea


Both the climatological and statistical approaches proposed in this study allowed us to
characterize the SCM dynamics in five regions of the Mediterranean Sea at large spatial
(interregional) and temporal (seasonal) scales. In the present section, we focused on the data
provided by two BGC-Argo floats that recorded simultaneously bio-optical properties, PAR
and nitrate concentration in two distinct regions, representing the two extremes of the
Mediterranean trophic gradient. This helped to gain understanding of the dynamics of the
SCM at a weekly and regional scale and should give insights in the mechanisms underlying
the occurrence of SCMs in these end-member regimes.

### 3.3.1 Overview of the two contrasted systems


The float WMO 6901512 (fGL) was been deployed in the Gulf of Lions the 11[th] of
April 2013 and recorded data until the 4[th] of May, 2014 (Figure 10a). The float WMO
6901528 (fLS) collected data in the Levantine Sea from May 18, 2013 to May 23, 2015
(Figure 10c). The two regions presented very different seasonal Chl*a* distribution. The Gulf of
Lions is a typical "temperate-like" system that exhibits a winter period characterized by large
MLDs (Millot, 1999; Lavigne et al., 2015) (maximum MLD > 1000 m, Figure 10d). The
intense mixing induces a refueling of nutrients (Gačić et al., 2002; D'Ortenzio et al., 2014;
Severin et al., 2017), which allows the development of a spring bloom (Marty et al., 2002,
2008; Mayot et al., 2017a) as revealed by the high surface Chl*a* from April to May (Figure
10b). A subsurface maximum of Chl*a* established from the end of May to mid-November at a
depth similar to that of the nitracline 1 µM and isolume 0.3 mol quanta m$^{-2}$ d$^{-1}$, and displayed
maximum Chl*a* of ~1 mg m$^{-3}$ in July (Figure 10b).
The Levantine Sea behaves, on the opposite, as a "tropical-like" system. Winter mixing
was weak (maximum MLD of 125 m; Figure 10d) but still able to erode the SCM as




suggested by the small increase in surface Chl$a$ from November to February (Figure 10b).
The seasonal MLD deepening almost never reached the nitracline, thus limiting the nitrate
supply to the upper layer of the water column (Dugdale and Wilkerson, 1988; Lavigne et al.,
2013; Pasqueron de Fommervault et al., 2015a), hence leading to relatively low surface
primary production in this area (Bricaud et al., 2002; Krom et al., 1991; Psarra et al., 2000;
Siokou-Frangou et al., 2010). The SCM is a permanent feature in this region, settling below
the isolume 0.3 mol quanta $m^{-2}$ $d^{-1}$ and far above the nitracline (Figure 10d).

### 3.3.2   Factors limiting the SCM

For exploring the limiting factors at the level of the SCM, we used a nutrient-vs-light
resource-limitation diagram. This approach employed in biogeochemical modelling (Cloern,
1999; Li and Hansell, 2016) exploits simultaneously PAR and $[NO_3^-]$ data in order to
understand which environmental factor limits phytoplankton growth (Figure 11).
In the Gulf of Lions, two different types of situations occurred: (1) very low light compared to
the maximum surface PAR ($PAR_{norm} < 0.025$) coupled with $NO_{3\ norm}^-$ comprised between 0
and 1, indicative of light limitation; and (2) low light compared to the maximum surface PAR
($PAR_{norm}$ within the range 0.025-0.15) associated with $NO_{3\ norm}^- < 0.15$, indicative of nitrate
limitation, probably resulting from uptake by phytoplankton (Figure 11a). On the contrary, in
the Eastern part of the Med Sea, the SCM was always associated with very low light
conditions compared to the maximum surface PAR ($PAR_{norm} < 0.025$) and variable $NO_{3\ norm}^-$
values comprised between 0.1 and 1 (Figure 11b). This suggests that, even when the nitrate
concentration is sufficient to sustain primary production at the SCM level, another factor
limits phytoplankton growth. Phytoplankton growth at the SCM is probably limited by light
or co-limited by both light and nutrients. Phosphate is also an important limiting factor for
phytoplankton growth in the entire Mediterranean Sea (Marty et al., 2002; Pujo-Pay et al.,



2011), the Eastern Basin in particular (Krom et al., 1991, 2010). Hence, in a non-nitrate
limited SCM of the Levantine (Figure 11b), phytoplankton may still be limited by either or
both low phosphate concentrations and low light levels. Since autonomous measurements of
phosphate concentrations are not possible yet, our chemical data are restricted to nitrate so we
cannot conclude on the role of phosphate in the settlement of the SCM.
The coupling between Chl$a$ and $b_{bp}$ was studied using the Chl$a$-to-$b_{bp}$ ratio. In both the
Western and Eastern Basins, SCMs with prevailing very low light conditions were
accompanied by high values of the Chl$a$-to-$b_{bp}$ ratio (> 300 mg m$^{-2}$). In contrast, in the SCM
of the Western Basin associated with low values of NO$_3^-$$_{norm}$, the Chl$a$-to-$b_{bp}$ ratio showed
values < 300 mg m$^{-2}$. This ratio is a proxy of the Chl$a$-to-POC ratio (Behrenfeld et al., 2015;
Álvarez et al., 2016; Westberry et al., 2016) and constitutes an optical index of
photoacclimation (Behrenfeld et al., 2005; Siegel et al., 2005) or of the phytoplankton
communities (Cetinić et al., 2012, 2015). Hence, in both the Western and Eastern Basins, the
high values of the Chl$a$-to-$b_{bp}$ ratio occurring in the SCM associated with very low light
conditions could be attributed to either photoacclimation of phytoplankton cells to low light
intensity. In contrast, in the SCM of the Western Basin where low values of NO$_3^-$$_{norm}$ were
reported, the low Chl$a$-to-$b_{bp}$ ratio values could either indicate a higher proportion of detrital
particles or an increase in biomass sustained by a specific phytoplankton assemblage
dominated by communities of nano- or pico-sized cells, including very small diatoms (*e.g.*
Leblanc et al., 2018).
**4 CONCLUSIONS**
The present study is, to our knowledge, the first examining the spatial and temporal
variability of Subsurface Chlorophyll *a* Maxima (SCMs) in the Mediterranean Sea using
BioGeochemical-Argo profiling floats equipped with both light (PAR) and nitrate ([NO$_3^-$])



sensors. Our study aims to improve the understanding of the characteristics and dynamics of
phytoplankton biomass in the subsurface layer of the Mediterranean Sea. We identified two
major mechanisms controlling the occurrence of SCMs, *i.e.* (1) SCMs arising from an actual
increase in carbon biomass at depth (or SBMs) and benefiting from both light and nutrient
resources; and (2) SCMs that stem from an increase in intracellular chlorophyll *a*
concentration as a result of photoacclimation to low light levels. In the temperate-like system
of the Western Mediterranean Sea, SBMs are recurrent whereas in the "subtropical-like"
system of the Eastern Mediterranean Sea, SCMs are, at a first order, representative of
photoacclimation process. Using a statistical classification of vertical profiles of Chl*a* and $b_{bp}$
collected over the entire Mediterranean, we have evidenced different intermediate SCM
situations that can be summarized as follows (Figure 12):
1)    The $SBM_{aZeu}$ is a Subsurface Biomass Maximum that settles above the euphotic zone in

the Northwestern Mediterranean Sea (NW). It is the thinnest (~40m) and shallowest

(~60 m) biomass maximum. It is also the most intense, probably because it benefits

from adequate light and nutrient resources.

2)    The $SBM_{bZeu}$ establishes below the euphotic zone in the NW. As well as the $SBMs$ of the

Southwestern Mediterranean Sea (SW) and Tyrrhenian Sea (TYR), the $SBM_{bZeu}$ is less

intense than the $SBM_{aZeu}$.

3)    The $SCM$ of the SW and TYR as well as the $SCM_{aZeu}$ (*i.e.* settling above the euphotic

depth) of the Ionian (ION) and Levantine (LEV) Seas are not biomass subsurface

maxima, but reflect Chl*a* maxima resulting from photoacclimation. Moving from the

SW to LEV region, the amplitude of the SCM decreases while its thickness increases.

4)    The $SCM_{bZeu}$ of the ION and LEV settle below the euphotic depth and are deeper (~95

602          m) than all the other subsurface maxima. They represent the oligotrophic end-member

type of subsurface maxima in the Med Sea. In these types of SCMs, phytoplankton



communities most probably establish deep in the water column, in order to reach the
nutrient resources. These communities are likely photoacclimated, and also possibly
photoadapted, to the low light conditions encountered at such depths. The
phytoplankton assemblage is likely composed of picophytoplankton (Casotti et al.,
2003; Siokou-Frangou et al., 2010), including the low-light adapted *Prochloroccoccus*
ecotype (Brunet et al., 2006; Garczarek et al., 2007).
In stratified oligotrophic ecosystems, the SCM phytoplankton species may settle
especially deep and adapt to the prevailing low-light levels in order to benefit from more
nutrients. On the contrary, when nitrates are not a limiting factor at the SCM level (*e.g.* in the
northwestern region after the bloom period), the SCM is only controlled by the amount of
light available at depth. In either case, light is a crucial forcing parameter that controls the
depth of the SCM. Consistently with previous studies conducted in other open ocean regions
(Longhurst and Glen Harrison, 1989; Furuya, 1990; Severin et al., 2017), the present work
suggests that shallower SCMs tend to display larger phytoplankton biomass than deeper
SCMs. In our study, these biomass maxima are characterized by a coupling of Chl$a$ and $b_{bp}$
that suggests an increase in carbon biomass. Finally, the present results indicate that SBMs
represent a frequent feature in the Med Sea, which contrasts with the idea that SCMs in
oligotrophic regions typically result from photoacclimation of phytoplankton cells. Thus, we
suggest that the contribution of SCMs to primary production, which may be substantial
although ignored by current satellite-based estimates, should be further investigated.
**AKNOWLEDGEMENTS**
This paper represents a contribution to the following research projects: remOcean
(funded by the European Research Council, grant 246777), NAOS (funded by the Agence
Nationale de la Recherche in the frame of the French ''Equipement d'avenir'' program, grant



ANR J11R107-F), the SOCLIM (Southern Ocean and climate) project supported by the
French research program LEFE- CYBER of INSU-CNRS, the Climate Initiative of the
foundation BNP Paribas and the French polar institute (IPEV), AtlantOS (funded by the
European Union's Horizon 2020 Research and Innovation program, grant 2014– 633211), E-
AIMS (funded by the European Commission's FP7 project, grant 312642), U.K. Bio-Argo
(funded by the British Natural Environment Research Council—NERC, grant NE/
L012855/1), REOPTIMIZE (funded by the European Union's Horizon 2020 Research and
Innovation program, Marie Skłodowska-Curie grant 706781), Argo-Italy (funded by the
Italian Ministry of Education, University and Research - MIUR), and the French Bio-Argo
program (BGC-Argo France; funded by CNES-TOSCA, LEFE Cyber, and GMMC). We
thank the PIs of several BGC-Argo floats missions and projects: Giorgio Dall'Olmo
(Plymouth Marine Laboratory, United Kingdom; E-AIMS and U.K. Bio- Argo); Kjell-Arne
Mork (Institute of Marine Research, Norway; E-AIMS); Violeta Slabakova (Bulgarian
Academy of Sciences, Bulgaria; E-AIMS); Emil Stanev (University of Oldenburg, Germany;
E-AIMS); Claire Lo Monaco (Laboratoire d'Océanographie et du Climat: Expérimentations et
Approches Numériques); Pierre-Marie Poulain (National Institute of Oceanography and
Experimental Geophysics, Italy; Argo- Italy); Sabrina Speich (Laboratoire de Météorologie
Dynamique, France; LEFE- GMMC); Virginie Thierry (Ifremer, France; LEFE-GMMC);
Pascal Conan (Observatoire Océanologique de Banyuls sur mer, France; LEFE-GMMC);
Laurent Coppola (Laboratoire d'Océanographie de Villefranche, France; LEFE-GMMC);
Anne Petrenko (Mediterranean Institute of Oceanography, France; LEFE-GMMC); and Jean-
Baptiste Sallée (Laboratoire d'Océanographie et du Climat, France; LEFE-GMMC). Louis
Prieur and Jean-Olivier Irisson (Laboratoire d'Océanographie de Villefranche, France) are
acknowledged for useful comments and fruitful discussion. We also thank the International





Argo Program and the CORIOLIS project that contribute to make the data freely and publicly
available.

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

**Figure captions**
**Figure 1**: Geographic location of the multi-variable vertical profiles collected by the BGC-Argo
profiling floats in the Mediterranean Sea. The boundaries of the regions considered in this study are
indicated by the black rectangles. NW, SW and TYR correspond to the Western Basin regions
whereas ION and LEV represent the Eastern Basin regions. The red color indicates BGC-Argo floats
equipped with nitrate sensors.
**Figure 2**: Comparison of the nitrate concentrations retrieved from the BGC-Argo floats to the
reference *in situ* measurements. The statistics ($R^2$ and slope) of the regression model between float
derived and *in situ* data are also indicated.
**Figure 3**: Boxplot of the distribution of the chlorophyll *a* concentration (Chl*a*) in the surface (a) and
SCM layers (b), the particulate backscattering coefficient ($b_{bp}$) in the SCM layer (c), and the depth (d)
and thickness (e) of the SCM for each Mediterranean region considered in this study.
**Figure 4**: Monthly median value of the chlorophyll *a* concentration, Chl*a* (in green) and of the
particulate backscattering coefficient, $b_{bp}$ (in blue) in the SCM layer for the five Mediterranean regions
considered in this study. The annual median of Chl*a* (0.28 mg m$^{-3}$) and $b_{bp}$ (5.8x10$^{-4}$ m$^{-1}$) calculated for
the SCM layer and over the entire Mediterranean Sea are indicated by the green and blue horizontal
lines, respectively. Note the different scales of the y-axes in panels a-e.
**Figure 5**: Monthly median values of the Subsurface Chl*a* Maximum (in green), the nitracline (in
black), the Subsurface $b_{bp}$ Maximum (in blue) and our reference isolume (in yellow) for the five
Mediterranean regions. The depth of the nitracline is not shown for the SW as there is no BGC-Argo
float equipped with a nitrate sensor for this region.
**Figure 6**: Boxplot of the distribution, for each of the Mediterranean regions considered in this study,
of the difference between the depths of the nitracline 1 μM and of the isolume 0.3 mol quanta m$^{-2}$ d$^{-1}$



(a), of the daily PAR in the SCM layer (b), and of the depth (c) and slope (d) of the nitracline. The SW
is not always represented as there is no BGC-Argo float equipped with a nitrate sensor in this region.
**Figure 7**: Normalized vertical profiles of the chlorophyll $a$ concentration (Chl$a$) (a, c, e, g, and i) and
particulate backscattering coefficient ($b_{bp}$) (b, d, f, h, and j) for each of the considered Mediterranean
regions. The Chl$a$ and $b_{bp}$ are normalized to their individual profile maximum value, Chl$a_{max}$ and
$b_{bpmax}$, respectively, while the depth is normalized to the euphotic depth ($Z_{eu}$). The color code indicates
the different types of profiles, namely the different shapes are the "*bloom*", "*mixed*", "*SBM*"
(Subsurface Biomass Maximum) with a distinction between the "*SBM$_{aZeu}$*" and the "*SBM$_{bZeu}$*" (for
SBM occurring above or below the euphotic depth, respectively), and the "*SCM*" (Subsurface
Chlorophyll Maximum) with a distinction between the "*SCM$_{aZeu}$*" and the "*SCM$_{bZeu}$*" (for SCM
occurring or below the euphotic depth, respectively).
**Figure 8:** Monthly occurrence of the different types of profiles shapes for each of the five considered
Mediterranean regions. The color code indicates the type of profiles shape, namely "*bloom*", "*mixed*",
"*SBM*" (Subsurface Biomass Maximum) with a distinction between the "*SBM$_{aZeu}$*" and the "*SBM$_{bZeu}$*"
(for SBM occurring above or below the euphotic depth, respectively), and the "*SCM*" (Subsurface
Chlorophyll Maximum) with a distinction between the "*SCM$_{aZeu}$*" and the "*SCM$_{bZeu}$*" (for SCM
occurring or below the euphotic depth, respectively).
**Figure 9**: Normalized vertical profiles of the chlorophyll $a$ concentration (Chl$a$) (a,c,e, and g) and
particulate backscattering coefficient ($b_{bp}$) (b,d,f, and h) for each shape type. The Chl$a$ and $b_{bp}$ are
normalized to their individual profile maximum value, Chl$a_{max}$ and $b_{bpmax}$, respectively, while the
depth is normalized to the euphotic depth ($Z_{eu}$). The color code and the type of lines indicate the
region of the Mediterranean Sea and the different shapes, respectively. The different shapes are the
"*bloom*", "*mixed*", "*SBM*" (Subsurface Biomass Maximum) with a distinction between the "*SBM$_{aZeu}$*"
and the "*SBM$_{bZeu}$*" (for SBM occurring above or below the euphotic depth, respectively), and the
"*SCM*" (Subsurface Chlorophyll Maximum) with a distinction between the "*SCM$_{aZeu}$*" and the
"*SCM$_{bZeu}$*" (for SCM occurring or below the euphotic depth, respectively). Note the different scales of
the x-axes.



**Figure 10**: Trajectory and Chl*a* time series of the fGL (a-b) and fLS (c-d). On panels b and d, the
white line shows the isolume 0.3 mol quanta m$^{-2}$ d$^{-1}$, the blue line indicates the Mixed Layer Depth
(MLD) and the black line the nitracline 1 μM.
**Figure 11**: Nutrient versus light resource-limitation diagram for the two BGC-Argo floats deployed in
the Gulf of Lions (a) and Levantine Sea (b). The color of the data points indicates the Chl*a*-to-$b_{bp}$ ratio
values. The x- and y-axes respectively represent the PAR and [NO$_3^-$] values normalized to the
maximum value calculated over the float lifetime in the layer extending from the surface to below the
SCM. Note that the plots show only data collected within the SCM layer, thus corresponding to low
normalized PAR values (i.e. under 25% of the maximum PAR).
**Figure 12**: Schematic representation of the different situations of SCMs in the Mediterranean Sea for
the five considered regions of the Mediterranean Sea along the west-to-east gradient.















**Table 1:** Regions with the corresponding abbreviation and number of available floats
and profiles represented in the Mediterranean BGC-Argo database used in the present study

| Region | Basin | Abbreviation | Number of profiles | Number of floats |
|---|---|---|---|---|
| Gulf of Lions and Ligurian Sea | Western | NW | 980 | 11 |
| Algero-provencal Basin | Western | SW | 540 | 5 |
| Tyrrhenian Sea | Western | TYR | 553 | 5 |
| Ionian Sea | Eastern | ION | 936 | 8 |
| Levantine Sea | Eastern | LEV | 1041 | 7 |
| Total | 2 | 5 | 4050 | 36 |










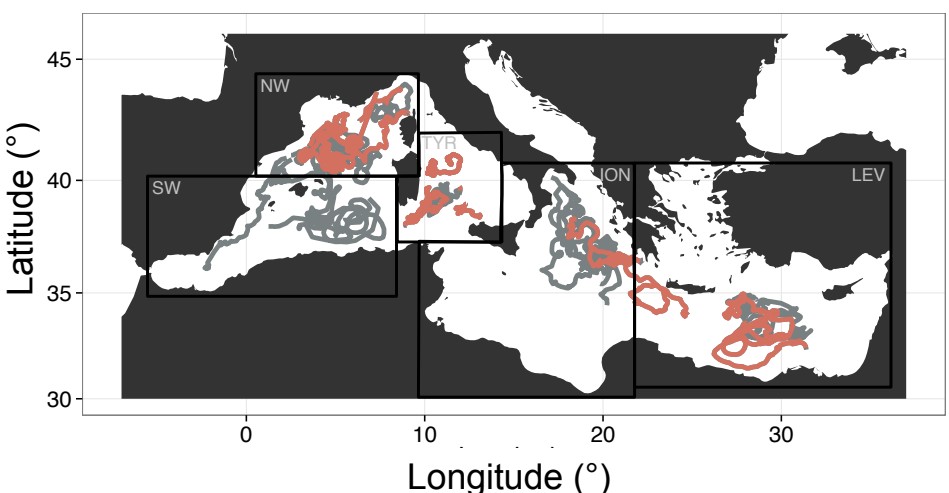


**Figure 1**: Geographic location of the multi-variable vertical profiles collected by the BGC-Argo

profiling floats in the Mediterranean Sea. The boundaries of the regions considered in this study are

indicated by the black rectangles. NW, SW and TYR correspond to the Western Basin regions

whereas ION and LEV represent the Eastern Basin regions. The red color indicates BGC-Argo floats

equipped with nitrate sensors.












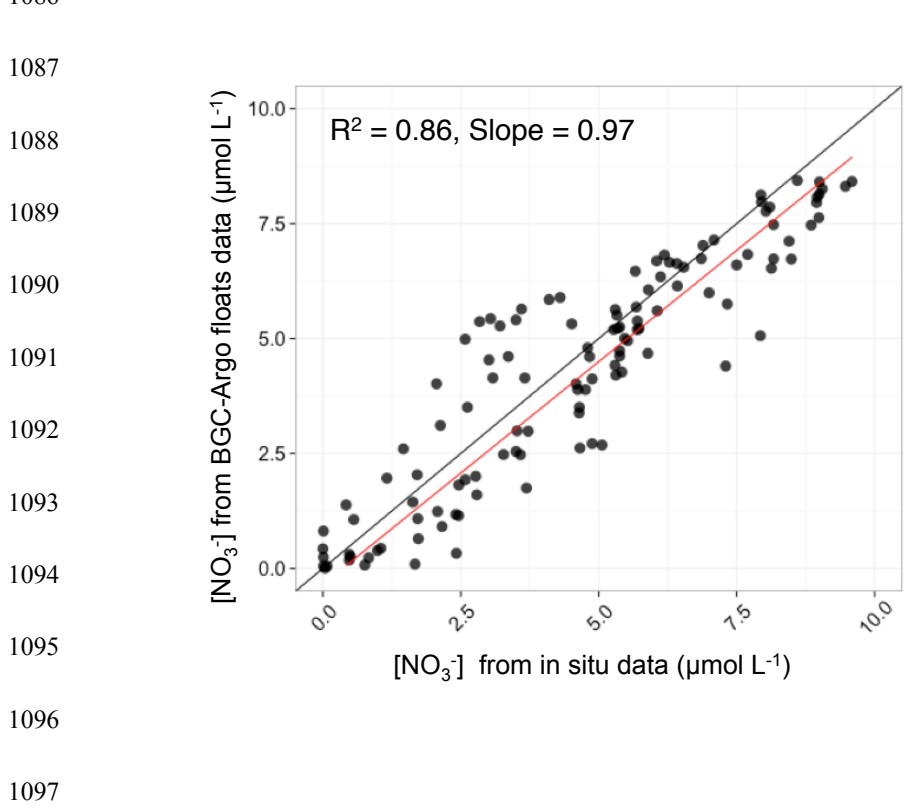

**Figure 2**: Comparison of the nitrate concentrations retrieved from the BGC-Argo floats to the reference *in situ* measurements. The statistics ($R^2$ and slope) of the regression model between float derived and *in situ* data are also indicated.






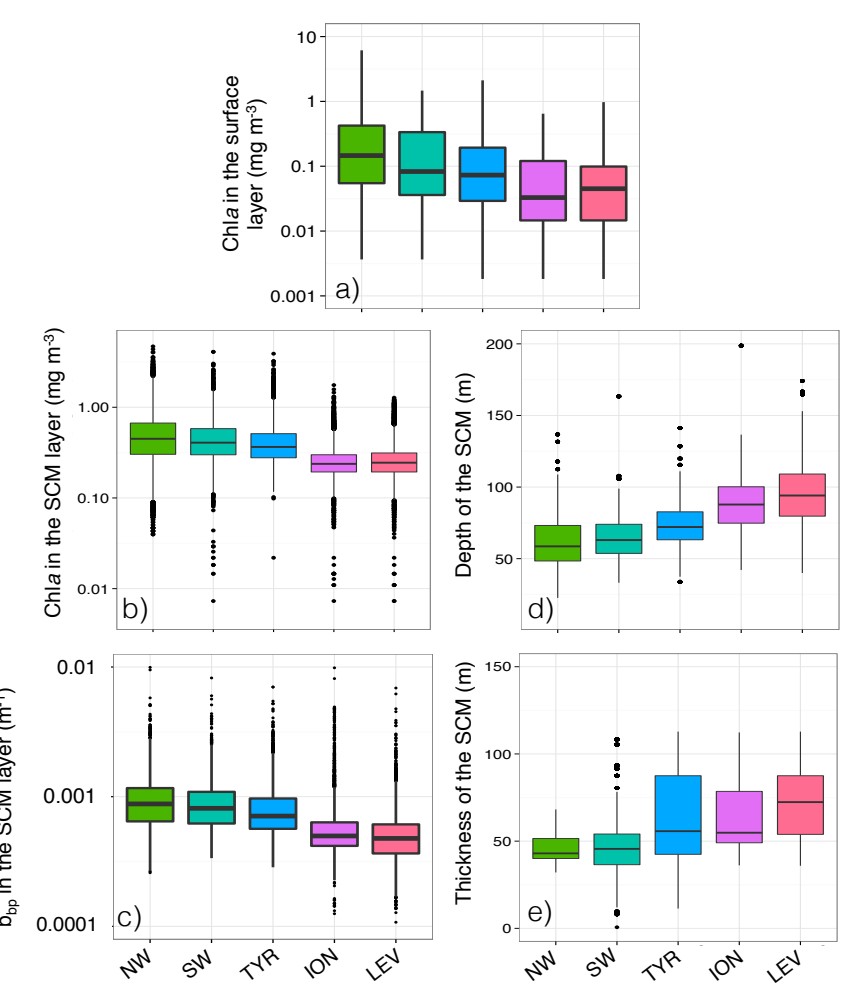

**Figure 3**: Boxplot of the distribution of the chlorophyll *a* concentration (Chl*a*) in the surface (a) and
SCM layers (b), the particulate backscattering coefficient ($b_{bp}$) in the SCM layer (c), and the depth (d)
and thickness (e) of the SCM for each Mediterranean region considered in this study.








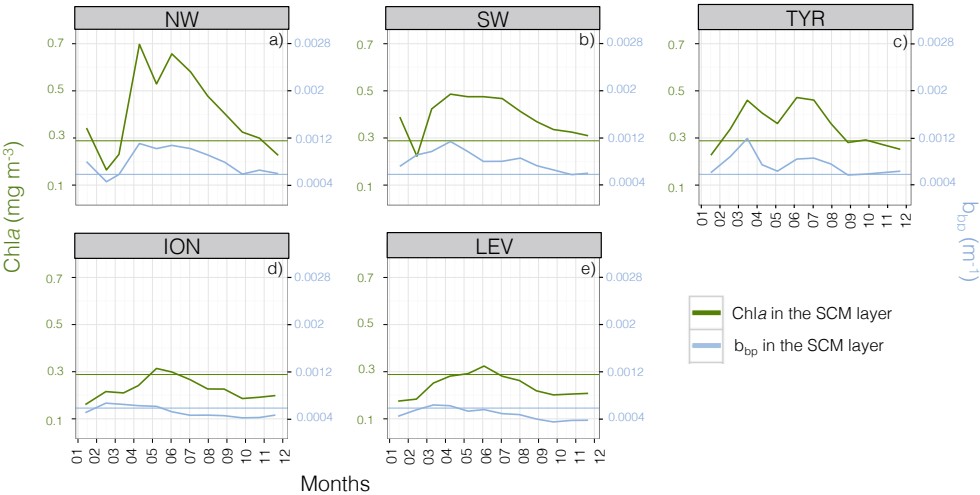


**Figure 4**: Monthly median value of the chlorophyll *a* concentration, Chl*a* (in green) and of the
particulate backscattering coefficient, $b_{bp}$ (in blue) in the SCM layer for the five Mediterranean regions
considered in this study. The annual median of Chl*a* (0.28 mg m$^{-3}$) and $b_{bp}$ (5.8x10$^{-4}$ m$^{-1}$) calculated for
the SCM layer and over the entire Mediterranean Sea are indicated by the green and blue horizontal
lines, respectively. Note the different scales of the y-axes in panels a-e.

















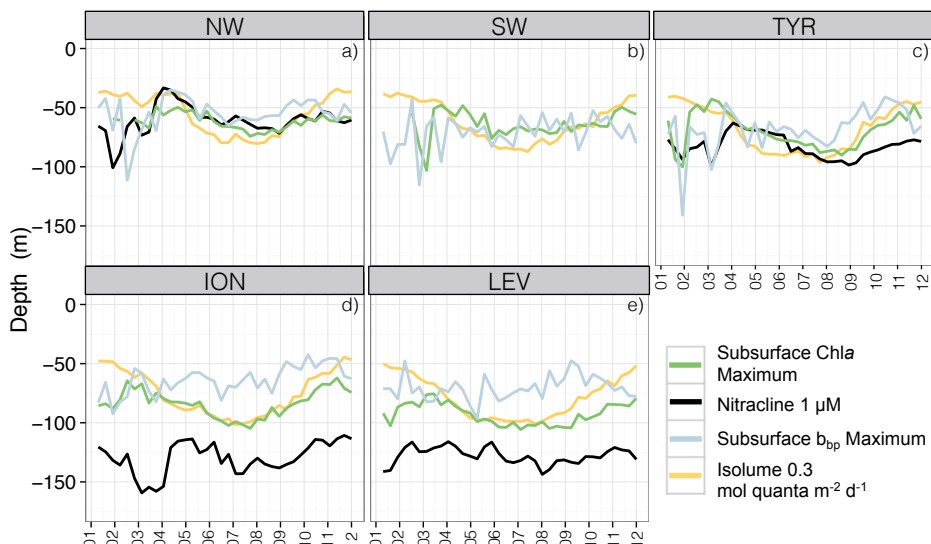


**Figure 5**: Monthly median values of the Subsurface Chl*a* Maximum (in green), the nitracline (in

black), the Subsurface $b_{bp}$ Maximum (in blue) and our reference isolume (in yellow) for the five

Mediterranean regions. The depth of the nitracline is not shown for the SW as there is no BGC-Argo

float equipped with a nitrate sensor for this region.












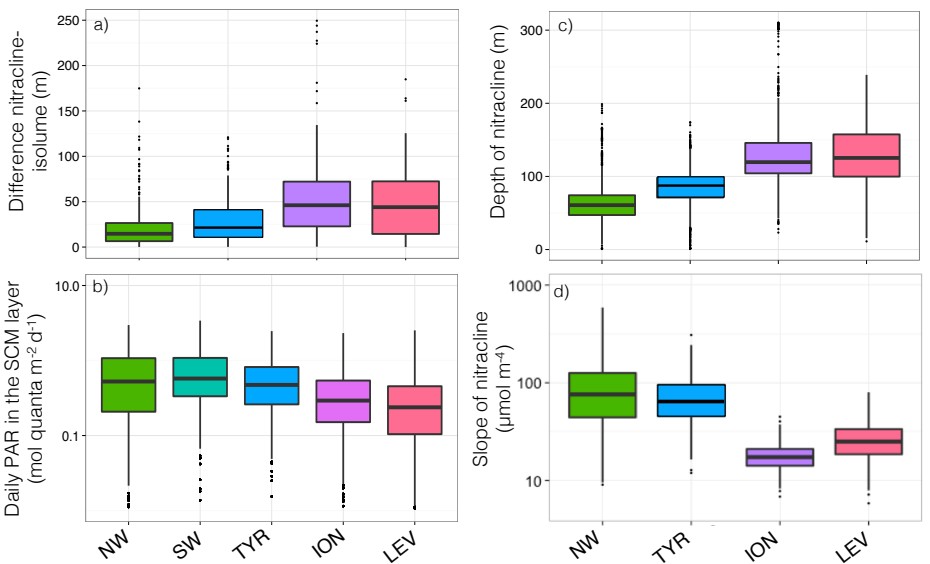

**Figure 6**: Boxplot of the distribution, for each of the Mediterranean regions considered in this study,
of the difference between the depths of the nitracline 1 µM and of the isolume 0.3 mol quanta m$^{-2}$ d$^{-1}$
(a), of the daily PAR in the SCM layer (b), and of the depth (c) and slope (d) of the nitracline. The SW
is not always represented as there is no BGC-Argo float equipped with a nitrate sensor in this region.
















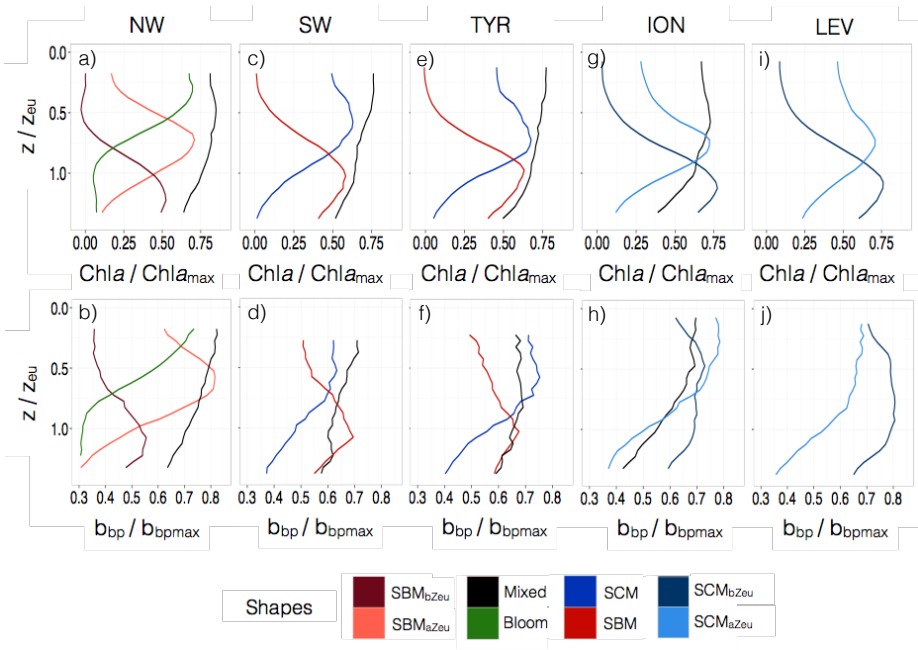


**Figure 7**: Normalized vertical profiles of the chlorophyll *a* concentration (Chl*a*) (a, c, e, g, and i) and
particulate backscattering coefficient ($b_{bp}$) (b, d, f, h, and j) for each of the considered Mediterranean
regions. The Chl*a* and $b_{bp}$ are normalized to their individual profile maximum value, Chl*a*$_{max}$ and
$b_{bpmax}$, respectively, while the depth is normalized to the euphotic depth ($Z_{eu}$). The color code indicates
the different types of profiles, namely the different shapes are the "*bloom*", "*mixed*", "*SBM*"
(Subsurface Biomass Maximum) with a distinction between the "*SBM$_{aZeu}$*" and the "*SBM$_{bZeu}$*" (for
SBM occurring above or below the euphotic depth, respectively), and the "*SCM*" (Subsurface
Chlorophyll Maximum) with a distinction between the "*SCM$_{aZeu}$*" and the "*SCM$_{bZeu}$*" (for SCM
occurring or below the euphotic depth, respectively).




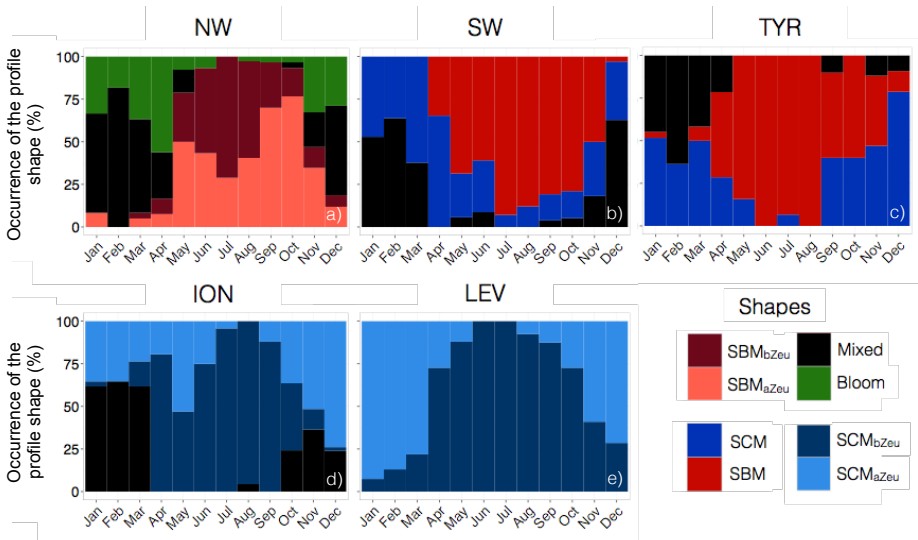


**Figure 8:** Monthly occurrence of the different types of profiles shapes for each of the five considered
Mediterranean regions. The color code indicates the type of profiles shape, namely "*bloom*", "*mixed*",
"*SBM*" (Subsurface Biomass Maximum) with a distinction between the "$SBM_{aZeu}$" and the "$SBM_{bZeu}$"
(for SBM occurring above or below the euphotic depth, respectively), and the "*SCM*" (Subsurface
Chlorophyll Maximum) with a distinction between the "$SCM_{aZeu}$" and the "$SCM_{bZeu}$" (for SCM
occurring or below the euphotic depth, respectively).











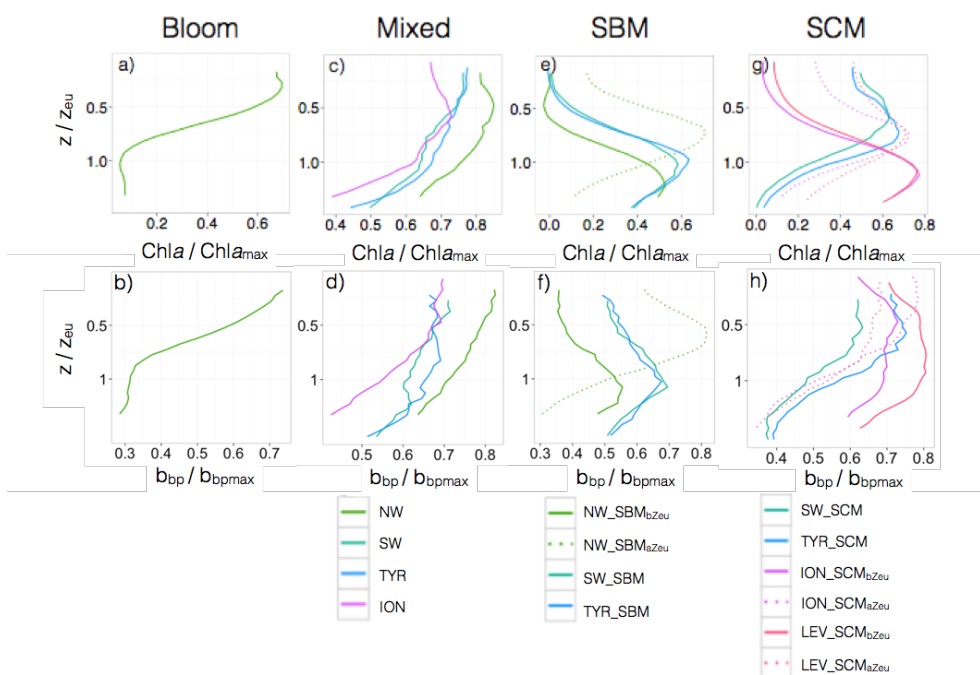


**Figure 9**: Normalized vertical profiles of the chlorophyll *a* concentration (Chl*a*) (a,c,e, and g) and particulate backscattering coefficient ($b_{bp}$) (b,d,f, and h) for each shape type. The Chl*a* and $b_{bp}$ are normalized to their individual profile maximum value, Chl*a*$_{max}$ and $b_{bpmax}$, respectively, while the depth is normalized to the euphotic depth ($Z_{eu}$). The color code and the type of lines indicate the region of the Mediterranean Sea and the different shapes, respectively. The different shapes are the "*bloom*", "*mixed*", "*SBM*" (Subsurface Biomass Maximum) with a distinction between the "*SBM$_{aZeu}$*" and the "*SBM$_{bZeu}$*" (for SBM occurring above or below the euphotic depth, respectively), and the "*SCM*" (Subsurface Chlorophyll Maximum) with a distinction between the "*SCM$_{aZeu}$*" and the "*SCM$_{bZeu}$*" (for SCM occurring or below the euphotic depth, respectively). Note the different scales of the x-axes.






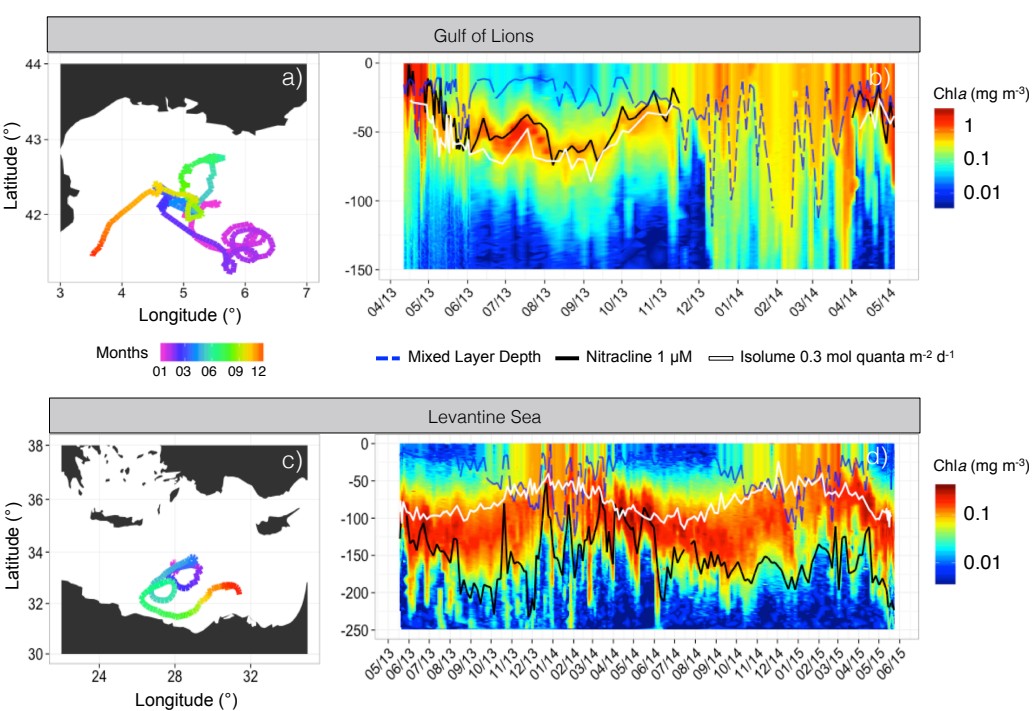


**Figure 10**: Trajectory and Chl*a* time series of the fGL (a-b) and fLS (c-d). On panels b and d, the

white line shows the isolume 0.3 mol quanta m$^{-2}$ d$^{-1}$, the blue line indicates the Mixed Layer Depth

(MLD) and the black line the nitracline 1 μM.











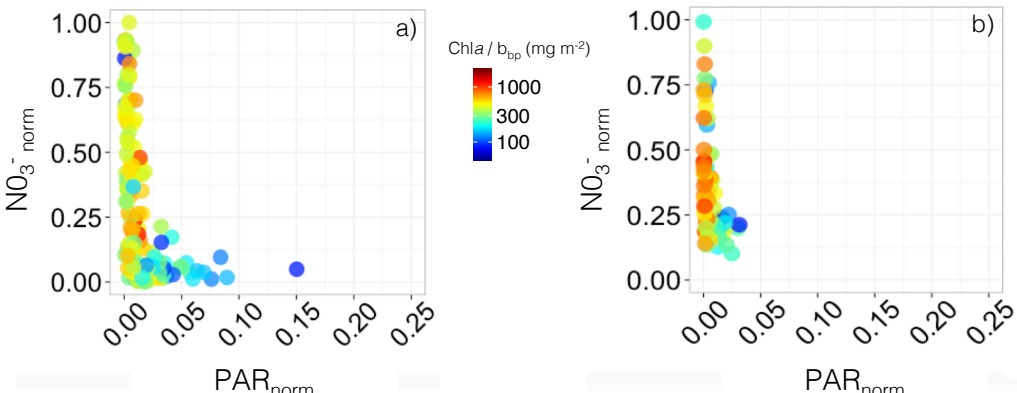


**Figure 11**: Nutrient versus light resource-limitation diagram for the two BGC-Argo floats deployed

in the Gulf of Lions (a) and Levantine Sea (b). The color of the data points indicates the Chl$a$-to-$b_{bp}$

ratio values. The x- and y-axes respectively represent the PAR and [$NO_3^-$] values normalized to the

maximum value calculated over the float lifetime in the layer extending from the surface to below the

SCM. Note that the plots show only data collected within the SCM layer, thus corresponding to low

normalized PAR values (i.e. under 25% of the maximum PAR).









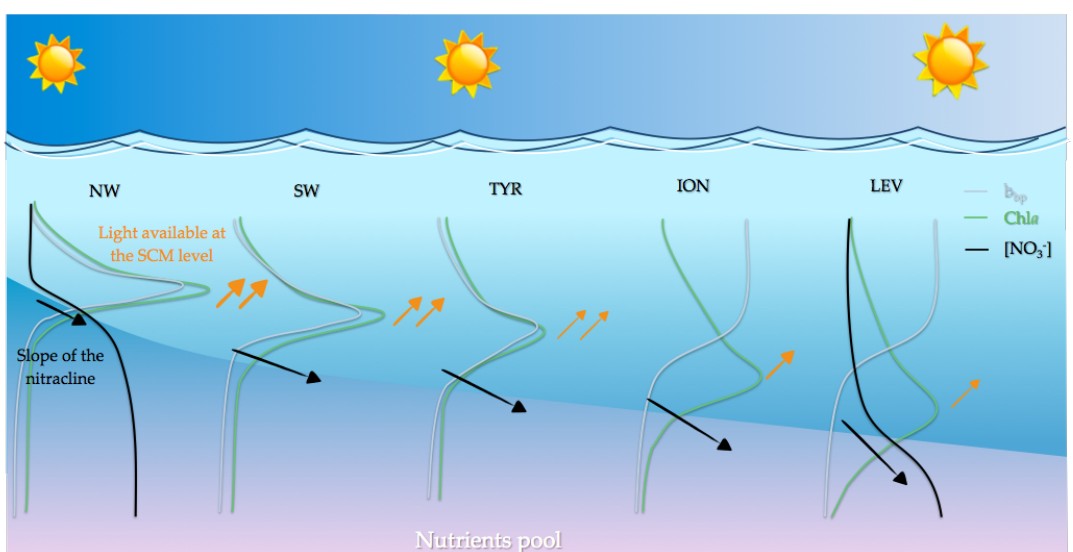


**Figure 12**: Schematic representation of the different situations of SCMs in the Mediterranean Sea for
the five considered regions of the Mediterranean Sea along the west-to-east gradient.


