# Peer review of "Bio-optical characterization of subsurface chlorophyll maxima in 1"

_Biogeosciences, 2018_

## Referee Comment (RC1) · Anonymous Referee #1 · 2 Nov 2018

Overview:

The paper presents novel dataset of Bio-Argo data collected in the Mediterranean Sea and assesses the occurrence of 'subsurface chlorophyll maxima' (SCM) in different regions and over time. In the more eutrophic regions the SCM, bbp maximum, nitracline and euphotic zone depth are generally well coupled, with the favourable nutrient and light availability at the SCM likely sustaining a phytoplankton biomass maximum. Conversely, in more oligotrophic conditions the SCM is often deeper than the backscatter maximum and the nitracline and euphotic zone depth are also decoupled, with the less

favourable light and nutrient availability at the SCM leading to SCM that are principally the result of photoacclimation. Phytoplankton within SCM are known to be ecologically and biogeochemically important and so understanding how these features vary is important and of interest to the readership of Biogeosciences. The paper also demonstrates the use of novel information provided by Bio-Argo, which is topical and timely. Although focussed on the Mediterranean Sea, the results inform on processes controlling SCM globally as well. Overall, I found the paper well written with clear aims and novel, informative results. I have only one general comment and provide specific comments below. Note that I was not able to judge the statistical methods used to classify the profiles (Appendix A), as this is outside my expertise.

General Comment:

Although SCMs have been assessed before in different regimes, a key strength of the paper is that a gradient across contrasting regimes is presented. The use of schematic diagram Fig 12 was very helpful to illustrate how the vertical profiles vary across this gradient. However, of key importance to driving the SCM dynamics across the different regimes is the physical forcing, and thus physical structure of the water column, and I felt the physical context was somewhat neglected in the data analysis and interpretations. Some comment is included in the later discussion (Fig 10) but I would urge the authors to consider adding a few sentences or short paragraph on the underlying physical controls in the different regimes and, in particular, consider adding the thermocline (or MLD?) to the schematic (Fig 12). It could also be useful to add to Fig 7 and/or 9 as well. Placing the observations into the physical context would provide a more complete explanation of the data presented and would help apply what is learnt to other regions globally.

Specific Comments:

- Line 29: suggest change "to understand which parameter controls the SCMs" to "to understand the main controls on the SCMs".

[Figure]

- Line 62: "their contributions to the depth integrated-production. . ... remains largely unknown. . .". The use of the Arctic example here is a bit of an odd choice, other examples could be added, for example the contribution is >40% in the oligotrophic Atlantic (Perez et al. 2006 Deep Sea Res 53:1616), 40-50% in the Celtic Sea (Hickman et al. 2012 MEPS 463:39); 58% in the North Sea (Weston et al. 2005 JPR 27:909). (The paper by Perez et al. is shows nicely the decoupling of Chl -a maxima, carbon maxima (.:. idea of Chl:C), thermocline, nitracline and 1% light depth in oligotrophic regimes that could be relevant to other statements about oligotrophic conditions as well).

- Lines 73-81. ". . ... Hence, this "miniature ocean" presents SCMs that may be encountered in both temperate environments and stratified waters of the global ocean". I found contrasting "temperate" and "oligotrophic" and/or "stratified" a bit confusing (many temperate regions are stratified), as it's not quite clear which properties of these different regimes are the relevant ones (seasonality? stratification? nutrient status? Maybe all of these?). Using 'seasonally stratified' vs 'permanently stratified' would be more precise?

- Methods Section: Please describe what (if any) correction for non-photochemical quenching was applied to the Chl-a fluorescence data.

- Line 146. Please give a reference for the quoted regional correction factors, or describe how they were obtained.

- Line 153: "0.03 kg m-3 density criterion", please describe what the criterion is.

- Section 2.3, 2.5. Im not familiar with the process of calibrating in situ nitrate sensors or the statistical tests applied, so cant make an informed judgements on these aspects. The methods seem reasonable to me.

- Line suggest changing "Occidental" and "Oriental" to a description more geo/oceanographic.

- Line 375: Suggest change "and presents an actual increase in phytoplankton biomass" to "that we propose supports an actual increase in phytoplankton biomass".

- Line 384: Suggest change "suggesting no accumulation of carbon at the SCM". It's unclear what you mean by accumulation here (implies sinking?), also carbon at a sub-surface bbp peak isn't necessarily accumulating. There is likely some generation and turnover of carbon at all depths in the water column, but the standing stock of biomass is maintained at a higher concentration at the depth of the bbp peak than the depth of the SCM. Suggest using more precise wording here.

- Line 393: please change "is, thus, limited by both the availability of light and nutrients" to "is, thus, likely to be limited by both light and nutrients". No measurements were made to assess whether phytoplankton were light or nutrient limited.

- Line 412-426. I found this section a little jumbled. The section on vertical species distributions and low light ecotypes seemed a bit out of place and it wasn't clear how it linked to the results presented. I suggest moving Lines 420-425 (which seem to provide the link) further up in this section, and re-consider the wording elsewhere to make the discussion easier to follow. The key points are there: that different phyto-plankton species or ecotypes are likely to have different depth and magnitude of C and Chl maxima, different Chl:C, and different bbp properties; gradients in taxa are likely (expected?) in stratified water columns, including through SCMs; and there are vertical gradients in the non-phytoplankton particles that contribute to bbp as well. Consequently, the overall Chl, C, and bbp profiles are the result of all taxa present, their bio-optical properties and their physiology, but it is not possible to tease these apart with the data. This is contained in the existing text, but could be clarified.

- Line 464: I suggest a very short description of what the "light driven hypothesis" is here.

- Line 581: "(1) SCMs arising from an actual increase in carbon biomass at depth (or SBMs) and benefitting from both light and nutrients". I think you have to be a little

careful here because the data didn't unequivocally show that phytoplankton biomass increased (the bbp max could be due to non-phytoplankton particles). Throughout the rest of the paper great care has been taken not to overinterpret bbp as phytoplankton carbon and to make careful statements about Chl-a:C (photoacclimation) with due consideration of non-phytoplankton particles contributing to the bbp signal. So, I suggest it's worth making sure this summary statement is equally precise. If by 'carbon biomass' you are being more general to include all plankton then say so, and distinguish from 'benefitting from both light and nutrients'.

- Throughout: the use of term "in the SCM layer" is often ambiguous as to whether you mean "at the SCM peak" or "integral within the SCM layer". For example, in the figure caption of Figure 3 and 4 it is not clear whether what's plotted is the Chl-a concentration at the SCM peak or an integrated Chl-a concentration through the SCM layer. The units (mg m-3) indicate the peak magnitude, but the words "in the SCM layer" imply the integral.

- Throughout: check that any abbreviations for Mediterranean Sea are used appropriately (Mediterranean Sea is used at the beginning but after a point "Med Sea" is used, e.g. Line 485).

---

## Referee Comment (RC2) · Anonymous Referee #2 · 6 Nov 2018

General Comment: The authors in this paper use number of Bio-Argo floats data to understand the distribution of Deep Chlorophyll maxima and its controlling environmental factors. The observation of biogeochemical parameters using modern autonomous floats is the need of the hour and they provide some critical information of BGC processes at higher spatial and temporal scale and the authors make use of a large data set from the Mediterranean basin. The technique and data used for this study is novel, interpretation of the data etc and the write up are all well organised and coherent. The authors have explained the quality control adequately that helps readers and other

researchers to plan for a similar observation and quality control of the data for other basins. I am aware of the fact is international BGC community is pushing hard to implement BIO-Argo on the scales of core Argo and this paper rightly justifies the need. The presence of deep chlorophyll maxima has been found and reported for many basins and authors have adequately cited their work. The new approach, however, is to use the slope or gradient to better understand the mechanism is an added advantage for this manuscript. I strongly recommend this manuscript for publication in Biogeosciences. Having said this, I have a few concerns which I thinks author should address. For example, though the authors have tried to explain the environmental factors that cause the presence of deep chlorophyll maxima, they have not explained the physical factors and their role. Much of their emphasis has been to relate the observations with parameters such as Par, Nitracline etc. I would like to include some description on the physical condition and variability in the MLD, thermocline etc. These factors also play a dominant role, particularly in defining the depth of nitracline or other nutrients distribution. The schematic explanation in figure 12 should also include the location (depth) of mixed layer and thermocline. I am not aware of the relation between thermocline and nitracline in Mediterranean sea but in tropical basins such as the Arabian sea, they are strongly coupled and one need to understand the variability in thermocline to understand the shape and depth of nitracline. The manuscript is well written; objectives and methods are well explained. Overall the manuscript reads well and is relatively easier to understand compared to the manuscript I usually receive for review.

---

## Author Comment (AC1) · 22 Jan 2019

We thank Reviewer #1 for his/her comments, which are displayed in blue font in this document. Our responses and description of any action taken in the revised manuscript follow each comment in black font.

1) Although SCMs have been assessed before in different regimes, a key strength of the paper is that a gradient across contrasting regimes is presented. The use of schematic diagram Fig12 was very helpful to illustrate how the vertical profiles vary

across this gradient. However, of key importance to driving the SCM dynamics across the different regimes is the physical forcing, and thus physical structure of the water column, and I felt the physical context was somewhat neglected in the data analysis and interpretations. Some comment is included in the later discussion (Fig 10) but I would urge the authors to consider adding a few sentences or short paragraph on the underlying physical controls in the different regimes and, in particular, consider adding the thermocline (or MLD?) to the schematic (Fig 12). It could also be useful to add to Fig 7 and/or 9 as well. Placing the observations into the physical context would provide a more complete explanation of the data presented and would help apply what is learnt to other regions globally.

Response: We agree with the Reviewer that the underlying physical controls of the SCM dynamics were not sufficiently considered in our analysis. Therefore we made substantial modifications to our manuscript in order to account for Reviewer #1 and #2's comments. We chose to consider the Mixed Layer Depth (MLD) as it seems to be a more complete indicator of the physical processes than the thermocline. In addition both the MLD and the thermocline had very similar temporal evolution hence we decided to represent only the MLD on the different figures for a better readability. We represented the value of the Mixed Layer Depth (MLD) on Figure 5, on Figure 7 and on the schematic representation of the different situations of SCMs in the Mediterranean Sea during the oligotrophic summer period shown in Figure 12. We also analysed the difference between the MLD and the nutricline depth and reported this information on Figure 6e. Our results indicate that the summer MLD exhibits very similar values among the considered regions and that, on the opposite, the winter MLD shows significantly different values between the Western and Eastern Basins. Hence, we suggest that the different mixing regimes and subsequent nutrient supply to the surface layer of the ocean may explain the seasonal succession and the amount of typical shapes of SCMs in the various regions of the Mediterranean Sea. For example, in the Northwestern region of the Mediterranean Sea, substantial mixing occurs during the winter period (MLD deeper than the nitracline) inducing a seasonal renewal of the nutrients available

in the surface and subsurface layers. In this region, 4 types of profiles of Chla and bbp are retrieved along the annual cycle and an SBM is observed during the oligotrophic period. On the opposite, in the Levantine Sea, the MLD is significantly shallower than the nitracline all year long, the upward diffusive flux of nitrates is weak and a SCM is systematically observed during the summer season.

To account for this comment, we modified the text in Section 3.1.3 (line numbers refer to the revised manuscript) as follows:

[revised manuscript text omitted]

2) Line 29: suggest change "to understand which parameter controls the SCMs" to "to understand the main controls on the SCMs".

Response: We modified the sentence as suggested by Reviewer #1: Âń Finally, a case study was performed on two contrasted regions and the environmental conditions at depth were further investigated to understand the main controls on the SCMs. Âż (p. 1 l. 27-29)

3) Line 62: "their contributions to the depth integrated-production [. . .] remains largely unknown [. . .]". The use of the Arctic example here is a bit of an odd choice, other examples could be added, for example the contribution is >40% in the oligotrophic Atlantic (Perez et al. 2006 Deep Sea Res 53:1616), 40-50% in the Celtic Sea (Hickman et al. 2012 MEPS 463:39); 58% in the North Sea (Weston et al. 2005 JPR 27:909). (The paper by Perez et al. is shows nicely the decoupling of Chl -a maxima, carbon maxima (idea of Chl:C), thermocline, nitracline and 1% light depth in oligotrophic regimes that could be relevant to other statements about oligotrophic conditions as well).

Response: Following the Reviewer's advice, we modified the sentence (line 62) giving examples of the underestimated production associated with the SCM in different regions of the global ocean: Âń Their contribution to the depth-integrated primary production has been assessed for a limited number of regions and remains largely unknown. It has been reported to be underestimated from 40 to 75% in the Arctic Ocean (Ardyna et al, 2013; Hill et al, 2013), to more than 40% in the oligotrophic Atlantic (Perez et al., 2006), 40-50% in the Celtic Sea (Hickman et al., 2012) and about 58% in the North Sea (Weston et al., 2005). Âż (p. 3 l. 60-65)

4) Lines 73-81. "[. . .] Hence, this "miniature ocean" presents SCMs that may be encountered in both temperate environments and stratified waters of the global ocean". I found contrasting "temperate" and "oligotrophic" and/or "stratified" a bit confusing (many temperate regions are stratified), as it's not quite clear which properties of these different regimes are the relevant ones (seasonality? stratification? nutrient status? Maybe all of these?). Using 'seasonally stratified' vs 'permanently stratified' would be more precise?

Response: We thank Reviewer #1 for this comment. Accordingly the terms 'seasonally stratified' vs. 'permanently stratified' are now used in the whole manuscript in order to avoid confusion: Âń Hence, this "miniature ocean" presents SCMs that may be encountered in both seasonally stratified environments and permanently stratified waters of the global ocean. Âż (p. 3 l. 82-83). We also modified the sentence: Âń In permanently stratified oligotrophic ecosystems, [. . .] Âż (p. 28 l. 666-667).

5) Methods Section: Please describe what (if any) correction for non-photochemical quenching was applied to the Chl-a fluorescence data.

Response: We added a sentence in Section 2.2 accordingly to the Reviewer's comment: Âń This procedure included a correction of non-photochemical quenching for Chla following Xing et al. (2012) method. Âż (p. 6 l. 142-143).

6) Line 146. Please give a reference for the quoted regional correction factors, or

describe how they were obtained.

Response: Following this comment and in order to better describe the Roesler et al. (2017) correction factor, we added the following sentences to Section 2.2:

Âń In addition, we applied a correction factor to Chla fluorescence measurements from the BGC-Argo floats, following the recommendation of Roesler et al. (2017). Comparing estimates of Chla from the WET Labs ECO fluorometers (used on BGC-Argo floats) with Chla estimates from other methods, these authors evidenced a bias varying according to the region sampled. In order to quantify this bias, they calculated the slope of the relationship between the Chla values from the ECO fluorometers and those estimated independently using HPLC analyses. This bias was further confirmed using optical proxies such as in situ radiometric measurements (Xing et al. 2011) or algal absorption measurements (Boss et al. 2013; Roesler and Barnard et al. 2013). At a global scale, Roesler et al. (2017) evidenced an overestimation of the Chla concentration by a factor of 2, on which regional variations of the fluorescence-to-Chla ratio are superimposed. Âż (p. 6 l. 143-154).

7) Line 153: "0.03 kg m-3 density criterion", please describe what the criterion is.

Response: In response to the Reviewer's comment we added the following sentence: Âń After binning the data at a 1-m resolution, the mixed layer depth (MLD) was derived from the CTD data using the density criterion of de Boyer Montégut (2004). The MLD was calculated as the depth where the density difference compared to the surface (10 m) reference value is 0.03 kg m-3.Âż (p. 7 l. 164-167).

8) Line suggest changing "Occidental" and "Oriental" to a description more geo/oceanographic.

Response: We used "western" and "eastern" instead of "Occidental" and "Oriental" according to the Reviewer's suggestion: Âń Similarly, the seasonal cycle of bbp in the SCM was more pronounced in the Western part of the Mediterranean Sea than in the

Eastern Basin. Âż (p. 15 l. 358-359).

9) Line 375: Suggest change "and presents an actual increase in phytoplankton biomass" to "that we propose supports an actual increase in phytoplankton biomass".

Response: We modified the sentence accordingly: Âń Hence, in the Western Basin of the Mediterranean Sea both light and nutrient resources seem to be available and probably support an actual increase in phytoplankton biomass (Figures 5 and 6a-b). Âż (p. 17, l. 408-410).

10) Line 384: Suggest change "suggesting no accumulation of carbon at the SCM". It's unclear what you mean by accumulation here (implies sinking?), also carbon at a subsurface bbp peak isn't necessarily accumulating. There is likely some generation and turnover of carbon at all depths in the water column, but the standing stock of biomass is maintained at a higher concentration at the depth of the bbp peak than the depth of the SCM. Suggest using more precise wording here.

Response: We agree with Reviewer #1 that the wording "accumulation of carbon" is not precise enough and, hence, modified the sentence accordingly: Âń The difference between the depths of the SCM and nitracline was ∼50 m during the stratified period (Figures 5d-e and 6a) and the SbbpM was shallower than the SCM (by ∼40 m), suggesting that the standing stock of carbon is maintained at a higher concentration above the depth of the SCM. Âż (p. 18 l. 418-422)

11) Line 393: please change "is, thus, limited by both the availability of light and nutrients" to "is, thus, likely to be limited by both light and nutrients". No measurements were made to assess whether phytoplankton were light or nutrient limited.

Response: We modified the sentence as follows: Âń The development of the SCM in this system is, thus, likely to be limited by the availability of both light and nutrients. Âż (p. 18 l. 432-434).

12) Line 412-426. I found this section a little jumbled. The section on vertical species

distributions and low light ecotypes seemed a bit out of place and it wasn't clear how it linked to the results presented. I suggest moving Lines 420-425 (which seem to provide the link) further up in this section, and re-consider the wording elsewhere to make the discussion easier to follow. The key points are there: that different phytoplankton species or ecotypes are likely to have different depth and magnitude of C and Chl maxima, different Chl:C, and different bbp properties; gradients in taxa are likely (expected?) in stratified water columns, including through SCMs; and there are vertical gradients in the non-phytoplankton particles that contribute to bbp as well. Consequently, the overall Chl, C, and bbp profiles are the result of all taxa present, their bio-optical properties and their physiology, but it is not possible to tease these apart with the data. This is contained in the existing text, but could be clarified.

Response: We modified Section 3.1.4 to clarify the discussion on this point:

Âń We have seen that the SCM of the Western Basin benefits from both light and nutrient resources. In these conditions, the observed simultaneous increase in Chla and bbp at the SCM most likely represents an actual development of phytoplankton biomass, as indicated by the concordance between the depths of the SCM and the SbbpM (Figure 5). On the opposite, in the Eastern part of the Mediterranean Sea, the maxima of Chla and bbp are not co-located. This result suggests that environmental conditions, typically the light conditions, might inhibit the increase in phytoplankton biomass.

In the Eastern Basin of the Mediterranean Sea, the microorganisms are, most probably, acclimated or even adapted to the environmental conditions. While photoacclimation is defined as a short-term acclimation of a photosynthetic organism to changing irradiance, photoadaptation refers to the long-term evolutionary adaptation of photosynthetic organisms to ambient light conditions, through genetic selection. SCM species are known to use different strategies such as photoacclimation to low light (i.e. increase in the intracellular pigment content), mixotrophy or small-scale directed movements towards light (Falkowski and Laroche, 1991; Geider et al., 1997; Clegg et al.,

2012). Phytoplankton species are also likely to have different carbon-to-chlorophyll ratio (Falkowski et al., 1985; Geider, 1987; Cloern et al., 1995; Sathyendranath et al., 2009) and bbp properties (Vaillancourt et al., 2004; Whitmire et al., 2010), and a vertical shift toward species photoadapted to the particular environmental conditions prevailing in the SCM layer is a well-known phenomenon (e.g. Pollehne et al., 1993; Latasa et al., 2016). For example, two ecotypes of Prochlorococcus, characterized by different accessory pigment contents are known to be adapted to either low-light or high-light conditions and to occupy different niches in the water column (Moore and Chisholm, 1999; Bouman et al., 2006; Garczarek et al., 2007). In particular, the low-light ecotype, characterized by increased intracellular pigmentation, has been frequently observed at the SCM level in the Mediterranean, especially in the Eastern part (Brunet et al., 2006; Siokou-Frangou et al., 2010). A west-to-east modification in the composition of phytoplankton communities in the SCM toward a dominance of picophytoplankton species adapted to recurring light limitation, has been observed (Christaki et al., 2001; Siokou-Frangou et al., 2010; Crombet et al., 2011). A vertical decoupling between bbp and Chla could thus illustrate either a photoacclimation of phytoplankton cells or the occurrence of specific phytoplankton communities adapted to the conditions prevailing in the SCM layer.

Although photoacclimation seems to be a widespread hypothesis in numerous studies to explain the vertical decoupling of Chla and bbp (e.g. Brunet et al., 2006; Cullen, 1982; Mignot et al., 2014), it should yet be reminded that this decoupling could also result from a change in the nature or size distribution of the entire particle pool. Small particles are, for example, known to backscatter light more efficiently than large particles (Morel and Bricaud, 1986; Stramski et al., 2004). A higher proportion of nonalgal particles in the Eastern compared to the Western Basin could thus explain the decoupling between bbp and Chla. The nonalgal particles compartment is defined as the background of submicronic living biological cells (i.e. viruses or bacteria) and non-living particles (i.e. detritus or inorganic particles) and is typically known to represent a significant part of the particulate assemblage in oligotrophic ecosystems (Morel and

Ahn, 1991; Claustre et al., 1999; Stramski et al., 2001).

Finally, photoacclimation processes as well as vertical gradients in phytoplankton species or in the non-phytoplankton particles, also contributing to bbp, could explain the vertical decoupling of bbp and Chla we observed in the Eastern Basin. The different types of Chla and bbp vertical profiles depends on both the nature of the particles present in the water column, the physiology of phytoplanktonic cells and their related bio-optical properties, but yet our dataset did not allow us to conclude on the dominance of one process compared to the other.Âż (p. 18-20 l. 437-485)

13) Line 464: I suggest a very short description of what the "light driven hypothesis" is here.

Response: We added a few sentences to address this comment: Âń These authors observed that the seasonal variation of the depth of the SCM depicts the same displacement as the isolumes and consequently suggested that the SCM depth displacement is light-driven. Âż (p. 22 l. 516-518)

14) Line 581: "(1) SCMs arising from an actual increase in carbon biomass at depth (or SBMs) and benefitting from both light and nutrients". I think you have to be a little careful here because the data didn't unequivocally show that phytoplankton biomass increased (the bbp max could be due to non-phytoplankton particles). Throughout the rest of the paper great care has been taken not to overinterpret bbp as phytoplankton carbon and to make careful statements about Chl-a:C (photoacclimation) with due consideration of non-phytoplankton particles contributing to the bbp signal. So, I suggest it's worth making sure this summary statement is equally precise. If by 'carbon biomass' you are being more general to include all plankton then say so, and distinguish from 'benefitting from both light and nutrients'.

Response: We thank Reviewer #1 for this comment and modified the sentence accordingly:

Âń (1) SCMs arising from an actual increase in carbon biomass most probably re-flecting an increase in phytoplankton biomass benefiting from both light and nutrient resources (SBMs) with a potentially non negligible contribution of non-phytoplankton particles at depth Âż (p. 26-27 l. 633-636).

15) Throughout: the use of term "in the SCM layer" is often ambiguous as to whether you mean "at the SCM peak" or "integral within the SCM layer". For example, in the figure caption of Figure 3 and 4 it is not clear whether what's plotted is the Chl-a con-centration at the SCM peak or an integrated Chl-a concentration through the SCM layer. The units (mg m-3) indicate the peak magnitude, but the words "in the SCM layer" imply the integral.

Response: To clarify the term "in the SCM layer", we added a section of explanation in Data and Methods:

Âń 2.5. Definition of the SCM Layer To study specifically the dynamics of the bio-optical properties in the SCM layer, we adjusted a Gaussian profile to each vertical profile of Chla of the database that presented a subsurface Chla maximum and computed the width of this SCM. This parameterizing approach proposed by Lewis et al. (1983) has been widely used to fit vertical profiles of Chla (e.g., Morel & Berthon, 1989; Uitz et al., 2006) such as:

$c(z) = c_{max} e^{(-(((z- z_{max})/\Delta z)^2))}$ (5)

where $c(z)$ is the Chla concentration at depth z, $c_{max}$ is the Chla concentration at the depth of the SCM ($z_{max}$), and $\Delta z$, the unknown, is the width of the SCM. In order to retrieve $\Delta z$, the unknown parameter, we performed an optimization of equation (5) with a maximum width set at 50 m so only the profiles with a relatively pronounced SCM are kept. Finally, in this study, the different biogeochemical variables are averaged in this SCM layer (cf. Figures 3, 4, 6 and 11)Âż (p. 12 l. 300-312).

16) Throughout: check that any abbreviations for Mediterranean Sea are used appropriately (Mediterranean Sea is used at the beginning but after a point "Med Sea" is used, e.g. Line 485).

Response: Throughout the text we replaced the abbreviation "Med Sea" by "Mediterranean Sea".

In addition to our responses to Reviewers #1 and #2, we modified Figure 1 for a better presentation of the BGC-Argo dataset. We underlined in black the trajectories of the BGC-Argo float of the Gulf of Lions and Levantine Sea that are used in Figure 10 and 11. We also modified the scale in Figure 9, 10 (a and c) and 11 for a better clarity. In Figure 9, we systematically adjusted the abscise axis between 0 and 0.8. In Figure 10, we modified the legend of the time scale of the float trajectories (Figure 10 a and c). In Figure 11, we adjusted the abscise axis from 0 to 0.16 (Figure 11a) and from 0 to 0.04 (Figure 11b).

---

## Author Comment (AC2) · 22 Jan 2019

We thank Reviewer #2 for his/her comments, our responses and description of any action taken in the revised manuscript follow each comment .

Having said this, I have a few concerns, which I thinks author should address. For example, though the authors have tried to explain the environmental factors that cause the presence of deep chlorophyll maxima, they have not explained the physical factors and their role. Much of their emphasis has been to relate the observations with parameters such as Par, Nitracline etc. I would like to include some description on the physical condition and variability in the MLD, thermocline etc. These factors also play a dominant role, particularly in defining the depth of nitracline or other nutrients distribution. The schematic explanation in figure 12 should also include the location (depth) of mixed layer and thermocline. I am not aware of the relation between thermocline and nitracline in Mediterranean sea but in tropical basins such as the Arabian sea, they are strongly coupled and one need to understand the variability in thermocline to understand the shape and depth of nitracline.

Response: We thank Reviewer #2 for this comment. We agree that the underlying physical controls of the SCMs are not extensively considered in our analysis. Therefore we made substantial modifications to our manuscript in order to account for Reviewer #1 and #2's comments. We chose to consider the Mixed Layer Depth (MLD) as it seems to be a more complete indicator of the physical processes than the thermocline. In addition both the MLD and the thermocline had very similar temporal evolution hence we decided to represent only the MLD on the different figures for a better readability. We represented the value of the Mixed Layer Depth (MLD) on Figure 5, on Figure 7 and on the schematic representation of the different situations of SCMs in the Mediterranean Sea during the oligotrophic summer period shown in Figure 12. We also analysed the difference between the MLD and the nutricline depth and reported this information on Figure 6e. Our results indicate that the summer MLD exhibits very similar values among the considered regions and that, on the opposite, the winter MLD shows significantly different values between the Western and Eastern Basins. Hence, we suggest that the different mixing regimes and subsequent nutrient supply to the surface layer of the ocean may explain the seasonal succession and the amount of typical shapes of SCMs in the various regions of the Mediterranean Sea. For example, in the Northwestern region of the Mediterranean Sea, substantial mixing occurs during the winter period (MLD deeper than the nitracline) inducing a seasonal renewal of the nutrients available in the surface and subsurface layers. In this region, 4 types of profiles of Chla and bbp are retrieved along the annual cycle and an SBM is observed during

the oligotrophic period. On the opposite, in the Levantine Sea, the MLD is significantly shallower than the nitracline all year long, the upward diffusive flux of nitrates is weak and a SCM is systematically observed during the summer season.

To account for this comment, we modified the text in Section 3.1.3 (line numbers refer to the revised manuscript) as follows:

[revised manuscript text omitted]

---

## Author Comment (AC3) · 22 Jan 2019

[revised manuscript text omitted]

Bethoux, J. P., P. Morin, C. Madec, and B. Gentili (1992), Phosphorus and nitrogen behaviour in the Mediterranean Sea, *Deep Sea Research Part A, Oceanographic Research Papers*, *39*(9), 1641–1654, doi:10.1016/0198-0149(92)90053-V.

Bosc, E., A. Bricaud, and D. Antoine (2004), Seasonal and interannual variability in algal biomass and primary production in the Mediterranean Sea, as derived from 4 years of SeaWiFS observations, *Global Biogeochemical Cycles*, *18*(GB1005), 1–17, doi:10.1029/2003GB002034.

Boss, E., M. Picheral, T. Leeuw, A. Chase, E. Karsenti, G. Gorsky, L. Taylor, W. Slade, J. Ras, and H. Claustre (2013), The characteristics of particulate absorption, scattering and attenuation coefficients in the surface ocean; Contribution of the Tara Oceans expedition, *Methods in Oceanography*, *7*, 52–62, doi:10.1016/j.mio.2013.11.002.

Bouman, H. et al. (2006), Oceanographic Basis of the Global Surface Distribution of Prochlorococcus Ecotypes, *Science*, *312*(5775), 918–921, doi:10.1126/science.39.1002.398.

de Boyer Montégut, C. (2004), Mixed layer depth over the global ocean: An examination of profile data and a profile-based climatology, *Journal of Geophysical Research*, *109*(C12), 1–20, doi:10.1029/2004JC002378.

Bricaud, A., E. Bosc, and D. Antoine (2002), Algal biomass and sea surface temperature in the Mediterranean Basin Intercomparison of data from various satellite sensors, and implications for primary production estimates, *Remote Sensing of Environment*, *81*(2–3), 163–178, doi:10.1016/S0034-4257(01)00335-2.

Briggs, N., M. J. Perry, I. Cetinić, C. Lee, E. D'Asaro, A. M. Gray, and E. Rehm (2011), High-resolution observations of
aggregate flux during a sub-polar North Atlantic spring bloom, *Deep Sea Research Part I: Oceanographic Research*
*Papers*, *58*(10), 1031–1039, doi:10.1016/j.dsr.2011.07.007.

Brunet, C., R. Casotti, V. Vantrepotte, F. Corato, and F. Conversano (2006), Picophytoplankton diversity and
photoacclimation in the Strait of Sicily (Mediterranean Sea) in summer. I. Mesoscale variations, *Aquatic Microbial*
*Ecology*, *44*(2), 127–141, doi:10.3354/ame044127.

Casotti, R., A. Landolfi, C. Brunet, F. D'Ortenzio, O. Mangoni, and M. Ribera d'Alcalá (2003), Composition and dynamics
of the phytoplankton of the Ionian Sea (eastern Mediterranean), *Journal of Geophysical Research*, *108*(C9), 1–19,
doi:10.1029/2002JC001541.

Cetinić, I., M. J. Perry, N. T. Briggs, E. Kallin, E. A. D'Asaro, and C. M. Lee (2012), Particulate organic carbon and inherent
optical properties during 2008 North Atlantic Bloom Experiment, *Journal of Geophysical Research*, *117*(C06028), 1–
18, doi:10.1029/2011JC007771.

Cetinić, I., M. J. Perry, E. D'Asaro, N. Briggs, N. Poulton, M. E. Sieracki, and C. M. Lee (2015), A simple optical index
shows spatial and temporal heterogeneity in phytoplankton community composition during the 2008 North Atlantic
Bloom Experiment, *Biogeosciences*, *12*(7), 2179–2194, doi:10.5194/bg-12-2179-2015.

Chiswell, S. M. (2011), Annual cycles and spring blooms in phytoplankton: Don't abandon Sverdrup completely, *Marine*
*Ecology Progress Series*, *443*, 39–50, doi:10.3354/meps09453.

Christaki, U., A. Giannakourou, F. Van Wambeke, and G. Grégori (2001), Nanoflagellate predation on auto- and
heterotrophic picoplankton in the oligotrophic Mediterranean Sea, *Journal of Plankton Research*, *23*(11), 1297–1310,
doi:10.1093/plankt/23.11.1297.

Claustre, H., A. Morel, M. Babin, C. Cailliau, D. Marie, J.-C. Marty, D. Tailliez, and D. Vaulot (1999), Variability in particle
attenuation and chlorophyll fluorescence in the tropical Pacific : Scales, patterns, and biogeochemical implications,
*Journal of Geophysical Research*, *104*(C2), 3401–3422.

Claustre, H. et al. (2010), Bio-optical profiling floats as new observational tools for biogeochemical and ecosystem studies:
Potential synergies with ocean color remote sensing., in *"Proceedings of the OceanObs'09: Sustained Ocean*
*Observations and Information for Society" Conference*, edited by J. Hall, D. E. Harrison, and D. Stammer, ESA Publ.
WPP-306, Venice, Italy, 21–25 Sep.

Clegg, M. R., U. Gaedke, B. Boehrer, and E. Spijkerman (2012), Complementary ecophysiological strategies combine to
facilitate survival in the hostile conditions of a deep chlorophyll maximum, *Oecologia*, *169*(3), 609–622,
doi:10.1007/s00442-011-2225-4.

Cleveland, J. S., M. J. Perry, D. A. Kiefer, and M. C. Talbot (1989), Maximal quantum yield of photosynthesis in the northwest Sargasso Sea., *Journal of Marine Research*, *47*(4), 869–886.

Cloern, J. E. (1999), The relative importance of light and nutrient limitation of phytoplankton growth: A simple index of
coastal ecosystem sensitivity to nutrient enrichment, *Aquatic Ecology*, *33*(1), 3–16, doi:10.1023/A:1009952125558.

Cloern, J. E., C. Grenz, and L. Videgar-Lucas (1995), An empirical model of the phytoplankton chlorophyll: carbon ration-
the conversion factor between productivity and growth rate., *Limnology and Oceanography*, *40*(7), 1313–1321.

Crombet, Y., K. Leblanc, B. Queguiner, T. Moutin, P. Rimmelin, J. Ras, H. Claustre, N. Leblond, L. Oriol, and M. Pujo-Pay
(2011), Deep silicon maxima in the stratified oligotrophic Mediterranean Sea, *Biogeosciences*, *8*(2), 459–475,
doi:10.5194/bg-8-459-2011.

Cullen, J. J. (1982), The Deep Chlorophyll Maximum: Comparing Vertical Profiles of Chlorophyll a, *Canadian Journal of*
*Fisheries and Aquatic Sciences*, *39*(5), 791–803, doi:10.1139/f82-108.

Cullen, J. J., and R. W. Eppley (1981), Chlorophyll Maximum Layers of the Southern-California Bight and Possible
Mechanisms of their Formation and Maintenance, *Oceanologica Acta*, *4*(1), 23–32.

D'Ortenzio, F., and M. R. D'Alcalà (2009), On the trophic regimes of the Mediterranean Sea: A satellite analysis,
*Biogeosciences*, *6*(2), 139–148, doi:10.5194/bg-6-139-2009.

D'Ortenzio, F. et al. (2014), Observing mixed layer depth, nitrate and chlorophyll concentrations in the northwestern
Mediterranean: Acombined satellite and NO3 profiling floats experiment, *Geophysical Research Letters*, *41*, 6443–
6451, doi:10.1002/2014GL061020.

Dall'Olmo, G., and K. A. Mork (2014), Carbon export by small particles in the Norwegian Sea, *Geophysical Research*
*Letters*, *41*(8), 2921–2927, doi:10.1002/2014GL059244.

Dubinsky, Z., and N. Stambler (2009), Photoacclimation processes in phytoplankton: Mechanisms, consequences, and
applications, *Aquatic Microbial Ecology*, *56*(2–3), 163–176, doi:10.3354/ame01345.

Dugdale, R. C., and F. P. Wilkerson (1988), Nutrient sources and primary production in the Eastern Mediterranean, in
*Oceanologica Acta*, edited by H. J. Minas and P. Nival, pp. 179–184.

Estrada, M., C. Marrasé, M. Latasa, E. Berdalet, M. Delgado, and T. Riera (1993), Variability of deep chlorophyll maximum
in the Northwestern Mediterranean, *Marine Ecology Progress Series*, *92*, 289–300, doi:10.3354/meps092289.

Falkowski, P. G., and J. Laroche (1991), Acclimation to spectral irradiance in algae, *Journal of Phycology*, *27*(1), 8–14,
doi:10.1111/j.0022-3646.1991.00008.x.

Falkowski, P. G., Z. Dubinsky, and K. Wyman (1985), Growth-irradiance relationships in phytoplankton, *Limnol. Oceanogr.*,
*30*(2), 311–321.

Fasham, M. J. R., T. Platt, B. Irwin, and K. Jones (1985), Factors affecting the spatial pattern of the deep chlorophyll maximum in the region of the Azores front, *Progress in Oceanography*, *14*(C), 129–165, doi:10.1016/0079-
6611(85)90009-6.

Fennel, K., and E. Boss (2003), Subsurface maxima of phytoplankton and chlorophyll: Steady-state solutions from a simple
model, *Limnology and Oceanography*, *48*(4), 1521–1534, doi:10.4319/lo.2003.48.4.1521.

Flory, E. N., P. S. Hill, T. G. Milligan, and J. Grant (2004), The relationship between floc area and backscatter during a
spring phytoplankton bloom, *Deep Sea Research Part I: Oceanographic Research Papers*, *51*(2), 213–223,
doi:10.1016/j.dsr.2003.09.012.

Furuya, K. (1990), Subsurface chlorophyll maximum in the tropical and subtropical western Pacific Ocean: Vertical profiles
of phytoplankton biomass and its relationship with chlorophyll a and particulate organic carbon, *Marine Biology*, *107*,
529–539, doi:10.1007/bf01313438.

Gačić, M., G. Civitarese, S. Miserocchi, V. Cardin, A. Crise, and E. Mauri (2002), The open-ocean convection in the
Southern Adriatic: A controlling mechanism of the spring phytoplankton bloom, *Continental Shelf Research*, *22*(14),
1897–1908, doi:10.1016/S0278-4343(02)00050-X.

Garczarek, L. et al. (2007), High vertical and low horizontal diversity of Prochlorococcus ecotypes in the Mediterranean Sea
in summer, *FEMS Microbiology Ecology*, *60*(2), 189–206, doi:10.1111/j.1574-6941.2007.00297.x.

Gardner, W. D., M. J. Richardson, and W. O. Smith (2000), Seasonal patterns of water column particulate organic carbon and
fluxes in the Ross Sea, Antarctica, *Deep Sea Research Part II: Topical Studies in Oceanography*, *47*, 3423–3449,
doi:10.1016/S0967-0645(00)00074-6.

Geider, R. J. (1987), Light and temperature dependence of the carbon to chlorophyll a ratio in microalgae and cyanobacteria:
Implications for physiology and growth of phytoplankton, *New Phytologist*, *106*(1), 1–34.

Geider, R. J. (1993), Quantitative phytoplankton physiology: implications for primary production and phytoplankton growth,
*ICES Marine Science Symposium*, *197*, 52–62.

Geider, R. J., H. L. MacIntyre, and T. M. Kana (1997), Dynamic model of phytoplankton growth and acclimation: Responses
of the balanced growth rate and the chlorophyll a:carbon ratio to light, nutrient-limitation and temperature, *Marine*
*Ecology Progress Series*, *148*(1–3), 187–200, doi:10.3354/meps148187.

Golub, G. H., and C. F. Van Loan (1996), *Matrix Computations*, The Johns., Baltimore and London.

Gong, X., W. Jiang, L. Wang, H. Gao, E. Boss, X. Yao, S. J. Kao, and J. Shi (2017), Analytical solution of the nitracline with
the evolution of subsurface chlorophyll maximum in stratified water columns, *Biogeosciences*, *14*(9), 2371–2386,
doi:10.5194/bg-14-2371-2017.

Gordon, H. R., and W. R. McCluney (1975), Estimation of the Depth of Sunlight Penetration in the Sea for Remote Sensing,

*Applied Optics*, *14*(2), 413–416, doi:10.1364/AO.14.000413.

Gutiérrez-Rodríguez, A., M. Latasa, M. Estrada, M. Vidal, and C. Marrasé (2010), Carbon fluxes through major
phytoplankton groups during the spring bloom and post-bloom in the Northwestern Mediterranean Sea, *Deep Sea*
*Research Part I: Oceanographic Research Papers*, *57*(4), 486–500, doi:10.1016/j.dsr.2009.12.013.

Hickman, A. E., C. M. Moore, J. Sharples, M. I. Lucas, G. H. Tilstone, V. Krivtsov, and P. M. Holligan (2012), Primary
production and nitrate uptake within the seasonal thermocline of a stratified shelf sea, *Marine Ecology Progress*
*Series*, *463*, 39–57, doi:10.3354/meps09836.

Hill, V. J., P. A. Matrai, E. Olson, S. Suttles, M. Steele, L. A. Codispoti, and R. C. Zimmerman (2013), Synthesis of
integrated primary production in the Arctic Ocean: II. In situ and remotely sensed estimates, *Progress in*
*Oceanography*, *110*, 107–125, doi:10.1016/j.pocean.2012.11.005.

Hodges, B. A., and D. L. Rudnick (2004), Simple models of steady deep maxima in chlorophyll and biomass, *Deep-Sea*
*Research Part I: Oceanographic Research Papers*, *51*(8), 999–1015, doi:10.1016/j.dsr.2004.02.009.

Holm-Hansen, O., and C. D. Hewes (2004), Deep chlorophyll-a maxima (DCMs) in Antarctic waters: I. Relationships
between DCMs and the physical, chemical, and optical conditions in the upper water column, *Polar Biology*, *27*(11),
699–710, doi:10.1007/s00300-004-0641-1.

Huot, Y., M. Babin, F. Bruyant, C. Grob, M. S. Twardowski, H. Claustre, and C. To (2007), Relationship between
photosynthetic parameters and different proxies of phytoplankton biomass in the subtropical ocean, *Biogeosciences*,
*4*(5), 853–868, doi:10.5194/bg-4-853-2007.

Ignatiades, L., S. Psarra, V. Zervakis, K. Pagou, E. Souvermezoglou, G. Assimakopoulou, and O. Gotsis-Skretas (2002),
Phytoplankton size-based dynamics in the Aegean Sea (Eastern Mediterranean), *Journal of Marine Systems*, *36*(1–2),
11–28, doi:10.1016/S0924-7963(02)00132-X.

Johnson, K., and H. Claustre (2016), Bringing Biogeochemistry into the Argo Age, *Eos*, 1–7, doi:10.1029/2016EO062427.

Johnson, K., W. Berelson, E. Boss, Z. Chase, H. Claustre, S. Emerson, N. Gruber, A. Körtzinger, M. J. Perry, and S. Riser
(2009), Observing Biogeochemical Cycles at Global Scales with Profiling Floats and Gliders: Prospects for a Global
Array, *Oceanography*, *22*(3), 216–225, doi:10.5670/oceanog.2009.81.

Johnson, K. S., and L. J. Coletti (2002), In situ ultraviolet spectrophotometry for high resolution and long-term monitoring of
nitrate, bromide and bisulfide in the ocean, *Deep-Sea Research Part I: Oceanographic Research Papers*, *49*(7), 1291–
1305, doi:10.1016/S0967-0637(02)00020-1.

Johnson, K. S. et al. (2017), Biogeochemical sensor performance in the SOCCOM profiling float array, *Journal of*
*Geophysical Research: Oceans*, *122*(8), 6416–6436, doi:10.1002/2017JC012838.

Kiefer, D. A., R. J. Olson, and O. Holm-Hansen (1976), Another look at the nitrite and chlorophyll maxima in the central
North Pacific, *Deep-Sea Research and Oceanographic Abstracts*, *23*(12), 1199–1208, doi:10.1016/0011-
7471(76)90895-0.

Kimor, B., T. Berman, and A. Schneller (1987), Phytoplankton assemblages in the deep chlorophyll maximum layers off the
Mediterranean coast of Israel, *Journal of Plankton Research*, *34*(11), 433–443, doi:10.1016/0198-0254(87)90913-7.

Klausmeier, C. a., and E. Litchman (2001), Algal games: The vertical distribution of phytoplankton in poorly mixed water
columns, *Limnology and Oceanography*, *46*(8), 1998–2007, doi:10.4319/lo.2001.46.8.1998.

Krom, M. D., N. Kress, S. Brenner, and L. I. Gordon (1991), Phosphorus Limitation of Primary Productivity in the Eastern
Mediterranean-Sea, *Limnology and Oceanography*, *36*(3), 424–432, doi:10.4319/lo.1991.36.3.0424.

Krom, M. D., K. C. Emeis, and P. Van Cappellen (2010), Why is the Eastern Mediterranean phosphorus limited?, *Progress
in Oceanography*, *85*(3–4), 236–244, doi:10.1016/j.pocean.2010.03.003.

Lacour, L., M. Ardyna, K. F. Stec, H. Claustre, L. Prieur, A. Poteau, M. Ribera D'Alcala, and D. Iudicone (2017),
Unexpected winter phytoplankton blooms in the North Atlantic subpolar gyre, *Nature Geoscience*, *10*(11), 836–839,
doi:10.1038/NGEO3035.

Latasa, M., A. Gutiérrez-rodríguez, A. M. Cabello, and R. Scharek (2016), Influence of light and nutrients on the vertical
distribution of marine phytoplankton groups in the deep chlorophyll maximum, *Planet Ocean*, *80*(S1), 57–62,
doi:10.3989/scimar.04316.01A.

Lavigne, H., F. D'Ortenzio, C. Migon, H. Claustre, P. Testor, M. R. D'Alcalà, R. Lavezza, L. Houpert, and L. Prieur (2013),
Enhancing the comprehension of mixed layer depth control on the Mediterranean phytoplankton phenology, *Journal of
Geophysical Research: Oceans*, *118*(7), 3416–3430, doi:10.1002/jgrc.20251.

Lavigne, H., F. D'Ortenzio, M. Ribera D'Alcalà, H. Claustre, R. Sauzède, and M. Gacic (2015), On the vertical distribution
of the chlorophyll a concentration in the Mediterranean Sea: a basin scale and seasonal approach, *Biogeosciences*,
*12*(5), 4139–4181, doi:10.5194/bgd-12-4139-2015.

Leblanc, K. et al. (2018), Nanoplanktonic diatoms are globally overlooked but play a role in spring blooms and carbon
export, *Nature Communications*, *9*(1), 953, doi:10.1038/s41467-018-03376-9.

Letelier, R. M., D. M. Karl, M. R. Abbott, and R. R. Bidigare (2004), Light driven seasonal patterns of chlorophyll and
nitrate in the lower euphotic zone of the North Pacific Subtropical Gyre, *Limnology and Oceanography*, *49*(2), 508–
519, doi:10.4319/lo.2004.49.2.0508.

Lewis, M. R., J. J. Cullen, and T. Platt (1983), Phytoplankton and thermal structure in the upper ocean: Consequences of
nonuniformity in chlorophyll profile, *Journal of Geophysical Research: Oceans*, *88*(C4), 2565–2570,
doi:10.1029/JC088iC04p02565.

Li, Q. P., and D. A. Hansell (2016), Mechanisms controlling vertical variability of subsurface chlorophyll maxima in a mode-
water eddy, *Journal of Marine Research*, *74*(3), 175–199, doi:10.1357/002224016819594827.

Loisel, H., and A. Morel (1998), Light scattering and chlorophyll concentration in case 1 waters: A reexamination,
*Limnology and Oceanography*, *43*(5), 847–858, doi:10.4319/lo.1998.43.5.0847.

Longhurst, A. R., and W. Glen Harrison (1989), The biological pump: Profiles of plankton production and consumption in
the upper ocean, *Progress in Oceanography*, *22*(1), 47–123, doi:10.1016/0079-6611(89)90010-4.

Marty, J. C., J. Chiavérini, M. D. Pizay, and B. Avril (2002), Seasonal and interannual dynamics of nutrients and
phytoplankton pigments in the western Mediterranean Sea at the DYFAMED time-series station (1991-1999), *Deep-
Sea Research Part II: Topical Studies in Oceanography*, *49*(11), 1965–1985, doi:10.1016/S0967-0645(02)00022-X.

Marty, J. C., N. Garcia, and P. Raimbault (2008), Phytoplankton dynamics and primary production under late summer
conditions in the NW Mediterranean Sea, *Deep-Sea Research Part I: Oceanographic Research Papers*, *55*(9), 1131–
1149, doi:10.1016/j.dsr.2008.05.001.

Mayot, N., F. D'Ortenzio, J. Uitz, B. Gentili, J. Ras, V. Vellucci, M. Golbol, D. Antoine, and H. Claustre (2017a), Influence
of the phytoplankton community structure on the spring and annual primary production in the Northwestern
Mediterranean Sea, *Journal of Geophysical Research: Oceans*, *122*, 1–17, doi:10.1002/2016JC012668.

Mayot, N., F. D'Ortenzio, V. Taillandier, L. Prieur, O. Pasqueron de Fommervault, H. Claustre, A. Bosse, P. Testor, and P.
Conan (2017b), Physical and biogeochemical controls of the phytoplankton blooms in North-Western Mediterranean
Sea: A multiplatform approach over a complete annual cycle (2012-2013 DEWEX experiment), *Journal of
Geophysical Research: Oceans*, *122*, doi:10.1002/2016JC012052.

Mignot, a., H. Claustre, F. D'Ortenzio, X. Xing, a. Poteau, and J. Ras (2011), From the shape of the vertical profile of in
vivo fluorescence to Chlorophyll-*a* concentration, *Biogeosciences*, *8*(8), 2391–2406, doi:10.5194/bg-8-2391-2011.

Mignot, A., H. Claustre, J. Uitz, A. Poteau, F. D'Ortenzio, and X. Xing (2014), Understanding the seasonal dynamics of
phytoplankton biomass and the deep chlorophyll maximum in oligotrophic environments: A Bio-Argo float
investigation, *Global Biogeochemical Cycles*, *28*(8), 1–21, doi:10.1002/2013GB004781.

Mignot, A., R. Ferrari, and H. Claustre (2018), Floats with bio-optical sensors reveal what processes trigger the North
Atlantic bloom, *Nature Communications*, *9*(1), 190, doi:10.1038/s41467-017-02143-6.

Mikaelyan, A. S., and G. A. Belyaeva (1995), Chlorophyll "a" content in cells of Antarctic phytoplankton, *Polar Biology*,
*15*(6), 437–445, doi:10.1007/BF00239721.

Millot, C. (1999), Circulation in the Western Mediterranean Sea, *Journal of Marine Systems*, *20*(1–4), 423–442,
doi:10.1016/S0924-7963(98)00078-5.

Moore, L. R., and S. W. Chisholm (1999), Photophysiology of the marine cyanobacterium Prochlorococcus: Ecotypic
differences among cultured isolates, *Limnology and Oceanography*, *44*(3), 628–638, doi:10.4319/lo.1999.44.3.0628.

Morel, A., and Y. Ahn (1991), Optics of heterotrophic nanoflagellates and ciliates: A tentative assessment of their scattering
role in oceanic waters compared to those of bacterial and algal cells, *Journal of Marine Research*, *49*(1), 177–202.

Morel, A., and J.-M. André (1991), Pigment distribution and Primary Production in the Western Mediterranean as Derived
and Modeled From Coastal Zone Color Scanner Observations, *Journal of Geophysical Research*, *96*(C7), 12685–
12698, doi:10.1029/91JC00788.

Morel, A., and J.-F. Berthon (1989), Surface pigments, algal biomass profiles, and potential production of the euphotic layer:
Relationships reinvestigated in view of remote-sensing applications, *Limnology and Oceanography*, *34*(8), 1545–1562,
doi:10.4319/lo.1989.34.8.1545.

Morel, A., and A. Bricaud (1986), Inherent optical properties of algal cells including picoplankton: theoretical and
experimental results, *Canadian Bulletin of Fisheries and Aquatic Science*, *214*, 521–559.

Morris, A. W., and J. P. Riley (1963), The determination of nitrate in sea water, *Analytica Chimica Acta*, *29*, 272–279,
doi:10.1016/S0003-2670(00)88614-6.

NREL (2000), SOLPOS 2.0 Documentation. Technical Report,

Organelli, E., H. Claustre, A. Bricaud, C. Schmechtig, A. Poteau, X. Xing, L. Prieur, F. D'Ortenzio, G. Dall'Olmo, and V.
Vellucci (2016), A novel near real-time quality-control procedure for radiometric profiles measured by Bio-Argo
floats: protocols and performances, *Journal of Atmospheric and Oceanic Technology*, *33*, 937–951,
doi:10.1175/JTECH-D-15-0193.1.

Organelli, E., H. Claustre, A. Bricaud, M. Barbieux, J. Uitz, F. D'Ortenzio, and G. Dall'Olmo (2017a), Bio-optical anomalies
in the world's oceans: An investigation on the diffuse attenuation coefficients for downward irradiance derived
fromBiogeochemical Argo float measurements, *Journal of Geophysical Research - Oceans*, *122*, 2017–2033,
doi:doi:10.1002/2016JC012629.

Organelli, E. et al. (2017b), Two databases derived from BGC-Argo float measurements for marine biogeochemical and bio-
optical applications, *Earth System Science Data*, *9*, 861–880, doi:https://doi.org/10.5194/essd-9-861-2017.

Parslow, J. S., P. W. Boyd, S. R. Rintoul, and F. B. Griffiths (2001), A persistent subsurface chlorophyll maximum in the
Interpolar Frontal Zone south of Australia: Seasonal progression and implications for phytoplankton-light-nutrient
interactions, *Journal of Geophysical Research: Oceans*, *106*(C12), 31543–31557, doi:10.1029/2000JC000322.

Pasqueron de Fommervault, O. et al. (2015a), Seasonal variability of nutrient concentrations in the Mediterranean Sea:
Contribution of Bio-Argo floats, *Journal of Geophysical Research: Oceans*, *120*, 8528–8550,
doi:doi:10.1002/2015JC011103.

Pasqueron de Fommervault, O., C. Migon, F. D′Ortenzio, M. Ribera d'Alcalà, and L. Coppola (2015b), Temporal variability of nutrient concentrations in the northwestern Mediterranean sea (DYFAMED time-series station), *Deep Sea Research*

*Part I: Oceanographic Research Papers*, *100*, 1–12, doi:10.1016/j.dsr.2015.02.006.

Pearson, K. (1901), On lines and planes of closest fit to systems of points in space, *Philosophical Magazine Series 6*, *2*(11),

559–572, doi:10.1080/14786440109462720.

Perez, V., E. Fernandez, E. Maranon, X. a. G. Moran, and M. V. Zubkov (2006), Vertical distribution of phytoplankton biomass, production and growth in the Atlantic subtropical gyres, *Deep Sea Res. I*, *53*, 1616–1634, doi:10.1016/j.dsr.2006.07.008.

Pollehne, F., B. Klein, and B. Zeitzschel (1993), Low light adaptation and export production in the deep chlorophyll maximum layer in the northern Indian Ocean, *Deep Sea Research Part II: Topical Studies in Oceanography*, *40*(3),

737–752, doi:10.1016/0967-0645(93)90055-R.

Psarra, S., a. Tselepides, and L. Ignatiades (2000), Primary productivity in the oligotrophic Cretan Sea (NE Mediterranean):

seasonal and interannual variability, *Progress in Oceanography*, *46*(2–4), 187–204, doi:10.1016/S0079-

6611(00)00018-5.

Quéguiner, B., P. Tréguer, I. Peeken, and R. Scharek (1997), Biogeochemical dynamics and the silicon cycle in the Atlantic sector of the Southern Ocean during austral spring 1992, *Deep-Sea Research Part II: Topical Studies in*

*Oceanography*, *44*(1–2), 69–89, doi:10.1016/S0967-0645(96)00066-5.

Raimbault, P., B. Coste, M. Boulhadid, and B. Boudjellal (1993), Origin of high phytoplankton concentration in deep chlorophyll maximum (DCM) in a frontal region of the Southwestern Mediterranean Sea (algerian current), *Deep-Sea*

*Research Part I*, *40*(4), 791–804, doi:10.1016/0967-0637(93)90072-B.

Roesler, C. et al. (2017), Recommendations for obtaining unbiased chlorophyll estimates from in situ chlorophyll fluorometers: A global analysis of WET Labs ECO sensors, *Limnology and Oceanography: Methods*, *15*(6), 572–585, doi:10.1002/lom3.10185.

Roesler, C. S., and A. H. Barnard (2013), Optical proxy for phytoplankton biomass in the absence of photophysiology:

Rethinking the absorption line height, *Methods in Oceanography*, *7*, 79–94, doi:10.1016/j.mio.2013.12.003.

Ryabov, A. B. (2012), Phytoplankton competition in deep biomass maximum, *Theoretical Ecology*, *5*(3), 373–385, doi:10.1007/s12080-012-0158-0.

Sakamoto, C. M., K. S. Johnson, and L. J. Coletti (2009), Improved algorithm for the computation of nitrate concentrations in seawater using an in situ ultraviolet spectrophotometer, *Limnology and Oceanography-Methods*, *7*, 132–143, doi:10.4319/lom.2009.7.132.

Sakamoto, C. M., K. S. Johnson, L. J. Coletti, and H. W. Jannasch (2017), Pressure correction for the computation of nitrate concentrations in seawater using an in situ ultraviolet spectrophotometer, *Limnology and Oceanography: Methods*,
*15*(10), 897–902, doi:10.1002/lom3.10209.

Sathyendranath, S., V. Stuart, A. Nair, K. Oka, T. Nakane, H. Bouman, M. H. Forget, H. Maass, and T. Platt (2009), Carbon-
to-chlorophyll ratio and growth rate of phytoplankton in the sea, *Marine Ecology Progress Series*, *383*, 73–84,
doi:10.3354/meps07998.

Schmechtig, C., A. Poteau, H. Claustre, F. D'Ortenzio, and E. Boss (2015), Processing Bio-Argo chlorophyll-a concentration
at the DAC Level, *Argo Data Management*, 1–22, doi:10.13155/39468.

Schmechtig, C., V. Thierry, and The Bio-Argo Team (2016a), Argo Quality Control Manual for Biogeochemical Data, *Argo*
*Data Management*, 1–54, doi:10.13155/40879.

Schmechtig, C., A. Poteau, H. Claustre, F. D'Ortenzio, G. Dall'Olmo, and E. Boss (2016b), Processing Bio-Argo particle
backscattering at the DAC level Version, *Argo Data Management*, 1–13, doi:doi:10.13155/39459.

Severin, T. et al. (2017), Open-ocean convection process: a driver of the winter nutrient supply and the spring phytoplankton
distribution in the Northwestern Mediterranean Sea, *Journal of Geophical Research*, doi:10.1002/2014JC010094.

Siegel, D. A., S. Maritorena, N. B. Nelson, and M. J. Behrenfeld (2005), Independence and interdependencies among global
ocean color properties: Reassessing the bio-optical assumption, *Journal of Geophysical Research C: Oceans*, *110*(7),
1–14, doi:10.1029/2004JC002527.

Siokou-Frangou, I., U. Christaki, M. G. Mazzocchi, M. Montresor, M. Ribera d'Alcalá, D. Vaqué, and A. Zingone (2010),
Plankton in the open Mediterranean Sea: a review, *Biogeosciences*, *7*(5), 1543–1586, doi:10.5194/bg-7-1543-2010.

Stramski, D., and D. A. Kiefer (1991), Light scattering by microorganisms in the open ocean, *Progress in Oceanography*,
*28*(4), 343–383, doi:10.1016/0079-6611(91)90032-H.

Stramski, D., R. A. Reynolds, M. Kahru, and B. G. Mitchell (1999), Estimation of Particulate Organic Carbon in the Ocean
from Satellite Remote Sensing, *Science*, *285*(5425), 239–242.

Stramski, D., A. Bricaud, and A. Morel (2001), Modeling the inherent optical properties of the ocean based on the detailed
composition of the planktonic community, *Applied Optics*, *40*(18), 2929–2945, doi:10.1364/AO.40.002929.

Stramski, D., E. Boss, D. Bogucki, and K. J. Voss (2004), The role of seawater constituents in light backscattering in the
ocean, *Progress in Oceanography*, *61*(1), 27–56, doi:10.1016/j.pocean.2004.07.001.

Taillandier, V. et al. (2017), Hydrography in the Mediterranean Sea during a cruise with RV Tethys 2 in May 2015, *Earth*
*System Science Data*, (November), 1–30, doi:10.17882/51678.

Tanhua, T., D. Hainbucher, K. Schroeder, V. Cardin, M. Álvarez, and G. Civitarese (2013), The Mediterranean Sea system: a
review and an introduction to the special issue, *Ocean Science*, *9*(5), 789–803, doi:10.5194/os-9-789-2013.

Tripathy, S. C., S. Pavithran, P. Sabu, H. U. K. Pillai, D. R. G. Dessai, and N. Anilkumar (2015), Deep chlorophyll maximum and primary productivity in Indian ocean sector of the southern ocean: Case study in the subtropical and polar front during austral summer 2011, *Deep Sea Research Part II: Topical Studies in Oceanography*, *118*, 240–249, doi:10.1016/j.dsr2.2015.01.004.

Uitz, J., H. Claustre, A. Morel, and S. B. Hooker (2006), Vertical distribution of phytoplankton communities in open ocean:

An assessment based on surface chlorophyll, *Journal of Geophysical Research*, *111*(C8005), 1–23, doi:10.1029/2005JC003207.

Uitz, J., H. Claustre, F. B. Griffiths, J. Ras, N. Garcia, and V. Sandroni (2009), A phytoplankton class-specific primary production model applied to the Kerguelen Islands region (Southern Ocean), *Deep Sea Research Part I:*

*Oceanographic Research Papers*, *56*(4), 541–560, doi:10.1016/j.dsr.2008.11.006.

Vaillancourt, R. D., C. W. Brown, R. R. L. Guillard, and W. M. Balch (2004), Light backscattering properties of marine phytoplankton: relationships to cell size, chemical composition and taxonomy, *Journal of Plankton Research*, *26*(2),

191–212, doi:10.1093/plankt/fbh012.

Videau, C., A. Sournia, L. Prieur, and M. Fiala (1994), Phytoplankton and primary production characteristics at selected sites in the geostrophic Almeria-Oran front system (SW Mediterranean Sea), *Journal of Marine Systems*, *5*(3–5), 235–250, doi:10.1016/0924-7963(94)90049-3.

Westberry, T. K., P. Schultz, M. J. Behrenfeld, J. P. Dunne, M. R. Hiscock, S. Maritorena, J. L. Sarmiento, and D. A. Siegel (2016), Annual cycles of phytoplankton biomass in the subarctic Atlantic and Pacific Ocean, *Global Biogeochemical*

*Cycles*, *30*(2), 175–190, doi:10.1002/2015GB005276.

Weston, K., L. Fernand, D. K. Mills, R. Delahunty, and J. Brown (2005), Primary production in the deep chlorophyll maximum of the central North Sea, *Journal of Plankton Research*, *27*(9), 909–922, doi:10.1093/plankt/fbi064.

Whitmire, A. L., W. S. Pegau, L. Karp-Boss, E. Boss, and T. J. Cowles (2010), Spectral backscattering properties of marine phytoplankton cultures, *Optics Express*, *18*(14), 15073–15093, doi:10.1029/2003RG000148.D.

Winn, C. D., L. Campbell, J. R. Christian, R. M. Letelier, D. V Hebel, J. E. Dore, L. Fujieki, and D. M. Karl (1995), 
[revised manuscript text omitted]